# EFFICODER: Enhancing Code Generation in Large Language Models through Efficiency-Aware Fine-tuning

**Dong Huang** [* 1 2]  **Guangtao Zeng** [* 3]  **Jianbo Dai** [4]  **Meng Luo** [2]  **Han Weng** [5]  **Yuhao Qing** [1]  **Heming Cui** [1]
**Zhijiang Guo** [† 6]  **Jie M. Zhang** [7]

## Abstract

As Large Language Models (LLMs) play an important role in code generation, enhancing both correctness and efficiency has become crucial. Current methods primarily focus on correctness, often overlooking efficiency. To address this gap, we introduce EFFICODER to improve both aspects by fine-tuning LLMs on a high-quality dataset EFFIINSTRUCT comprising correct and efficient code samples. Our method involves leveraging multiple LLMs to generate diverse candidate code solutions for various tasks across different programming languages. We then evaluate these solutions by measuring their execution time and memory usage through local execution. The code solution with the lowest execution time and memory consumption is selected as the final output for each task. Experimental results demonstrate significant improvements when fine-tuning with EFFIINSTRUCT. For instance, Qwen2.5-Coder-7B-Instruct's pass@1 score increases from 44.8% to 57.7%, while the average execution time for correct tasks decreases by 48.4%. EFFICODER offers a scalable and effective solution for advancing AI-driven code generation, benefiting both software development and computational problem-solving. Dataset and Code are available at https://github.com/huangd1999/EffiCoder.

## 1. Introduction

Large Language Models (LLMs) have recently made significant strides across various tasks (OpenAI, 2023; Anthropic, 2024; Meta, 2024), including code-related applications like code completion (Chen et al., 2021; Austin et al., 2021), debugging (Haque et al., 2022; Chen et al., 2023), and translation (Rozière et al., 2020; Ahmad et al., 2023). Before deploying LLMs into integrated development environments, it is crucial to ensure that the generated code meets the required efficacy standards. To address this, researchers have explored various datasets to fine-tune LLMs, thereby improving the efficacy of LLM-generated code (Ouyang et al., 2022; Wei et al., 2022). For example, Code Alpaca (Chaudhary, 2023) utilized the Self-Instruct framework (Wang et al., 2023) to synthesize data, while WizardCoder (Luo et al., 2024) employed the Evol-Instruct technique (Xu et al., 2024) to generate heuristic prompts for diverse solutions. Additionally, OSS-Instruct (Wei et al., 2024b) created new coding problems using open-source snippets with LLMs, and Octopack (Muennighoff et al., 2024) focused on curating high-quality Git commit messages that resemble natural language instructions. These efforts have led to increased correctness in LLM-generated code.

However, existing works primarily focus on enhancing the correctness of LLM-generated code while neglecting to optimize its efficiency. As a result, the efficiency of such code often falls short compared to canonical solutions written by human developers. Recent studies (Shi et al., 2024; Niu et al., 2024; Du et al., 2024; Huang et al., 2024a; Liu et al., 2024b) also point out that LLM-generated code typically exhibits lower efficiency in execution time and memory usage. For instance, on the EffiBench benchmark (Huang et al., 2024b), even the most advanced LLMs, such as GPT-4-Turbo, produced less efficient code, with average and worst-case execution times being 1.69 and 45.49 times longer than those of canonical solutions, respectively. Efficiency is crucial because inefficient code consumes more computational resources, leading to higher energy consumption and increased operational costs. This is particularly important in the context of sustainability, as the demand for computing power continues to grow, and reducing the environmental impact of large-scale computations becomes a pressing concern. Furthermore, inefficient code may be impractical for use in resource-constrained environments, such as mobile devices or embedded systems, where both energy and

---

[*]Equal contribution  [1]University of Hong Kong [2]National University of Singapore [3]Singapore University of Technology and Design [4]University of Edinburgh [5]Beijing University of Posts and Telecommunications [6]University of Cambridge [7]King's College London. Correspondence to: Zhijiang Guo <zg283@cam.ac.uk>.

*Proceedings of the 42nd International Conference on Machine Learning*, Vancouver, Canada. PMLR 267, 2025. Copyright 2025 by the author(s).

processing power are limited. This underscores the urgent need to develop new methods that can enhance both the **correctness** and **efficiency** of LLM-generated code.

In this paper, we introduce EFFICODER, aimed at fine-tuning LLMs to improve both code efficiency and correctness. We begin by aggregating source code from existing open-source datasets. This is followed by a rigorous preprocessing and cleaning process, coupled with the generation of test cases for each task to evaluate code efficiency. We leverage multiple LLMs to generate diverse candidate code solutions for various tasks across different programming languages. We then evaluate these solutions by directly measuring their execution time and memory usage through local execution. Code solutions with the lowest execution time and memory consumption are selected as the final output. The resulting optimized code, along with its associated metadata, forms EFFIINSTRUCT, which serves as a high-quality resource for training LLMs.

Extensive experiments demonstrate that fine-tuning LLMs with EFFIINSTRUCT improves both correctness and efficiency. For example, the fine-tuned Qwen2.5-Coder-7B-Instruct (Hui et al., 2024) increases the pass@1 from 44.8% and 76.2% to 57.7% and 78.0% on EffiBench and HumanEvalPlus, while also reducing the average execution time from 0.31 seconds to 0.16 seconds — representing a 48.4% reduction in execution time overhead on EffiBench. Compared to PIE (Shypula et al., 2024), which increases the pass@1 from 12.2% to 19.5% on HumanEvalPlus, the pass@1 of CodeLlama-7B (Rozière et al., 2023) fine-tuned with EFFIINSTRUCT further increases to 31.1%. In addition, EFFICODER decreases the execution time by 46.2% while PIE decreases it by 23.1%. We will fully open-source EFFICODER and the source code to facilitate research. To conclude, this paper makes the contributions:

- We propose a versatile framework for constructing code generation datasets with efficient solutions, adaptable to various programming languages and sources.

- We construct EFFIINSTRUCT, to the best of our knowledge, it is the first instruction-tuning dataset designed to improve the efficiency of LLM-generated code, facilitating fine-tuning for more efficient code generation.

- We introduce EFFICODER by fine-tuning various widely used LLMs using EFFIINSTRUCT, demonstrating improvements in both correctness and efficiency.

## 2. Related Works

### 2.1. LLMs for Code

The increasing popularity of LLMs for code generation has coincided with the growing availability of open-source code repositories and the need to boost developer productivity (Sun et al., 2024). Initial efforts focused on training models specifically for coding tasks, such as CodeT5 (Wang et al., 2021), AlphaCode (Li et al., 2022), CodeGen (Nijkamp et al., 2023), InCoder (Fried et al., 2023), StarCoder (Li et al., 2023a), SantaCoder (Allal et al., 2023), and DeepSeek-Coder (DeepSeekAI, 2023). Contrastingly, models such as Codex (Chen et al., 2021) and CodeLlama (Rozière et al., 2023) represent a subsequent stride, being fine-tuned from foundation models (Brown et al., 2020; Touvron et al., 2023). These code LLMs have been applied to various tasks, including code generation (Chen et al., 2021; Dai et al., 2024), program repair (Haque et al., 2022; Jiang et al., 2023), automated testing (Lemieux et al., 2023; Deng et al., 2023), code translation (Rozière et al., 2020; Ahmad et al., 2023), type prediction (Mir et al., 2022; Wei et al., 2023), and code summarization (Hasan et al., 2021; Ahmed & Devanbu, 2022). While LLMs have achieved impressive results in code generation tasks like HumanEval (Chen et al., 2021) and MHPP (Dai et al., 2024), their efficiency has received less attention. Recent studies (Shi et al., 2024; Huang et al., 2024b; Niu et al., 2024) have shown that LLM-generated code exhibits lower efficiency in terms of execution time and memory usage compared to canonical solutions. These findings highlight the need for further research and development to improve the efficiency of LLM-generated code. In this work, we propose the first fine-tuning method that significantly improves both the efficiency and correctness of code generated by various LLMs.

### 2.2. Instruction Tuning for Code

Instruction tuning has proven effective in enhancing the usability and overall performance of LLMs across various language tasks (Ouyang et al., 2022; Wei et al., 2022; Zhao et al., 2024). This approach has been extended to the domain of code generation. The core challenge is the acquisition of high-quality instructional data, which is often labor-intensive. To address this, recent research has focused on developing methods to generate synthetic instruction data. Studies have shown that textbook-quality synthetic data alone can improve a model's coding and reasoning capabilities (Gunasekar et al., 2023; Li et al., 2023b). One early effort was Self-Instruct (Wang et al., 2023), which utilized LLMs to generate synthetic instruction-response pairs using carefully crafted prompts. The same LLM was then instruction-tuned on this synthetic data. Code Alpaca (Chaudhary, 2023) applied the Self-Instruct approach with GPT models, tailoring it specifically for code generation, editing, and optimization tasks. Building upon this, WizardCoder (Luo et al., 2024) adapted the Evol-Instruct technique (Xu et al., 2024) to the coding domain by designing heuristic prompts to create more complex and diverse synthetic data. OSS-Instruct (Wei et al., 2024b) took a

Figure 1. Overview of the construction pipeline for EFFIINSTRUCT: We begin by collecting the initial EFFIINSTRUCT from different open-source datasets. Starting with the original code, we require multiple LLMs to generate candidate solutions, using test cases to profile execution overhead, and use the most efficient solution generated by LLMs as the solution for each task. We then have our final fine-tuning dataset, EFFIINSTRUCT, which consists of optimized code and rich metadata, designed to train models for generating efficient code.

Figure 2. Examples of code with varying efficiency levels: The first solution has high memory usage and long execution time. The second reduces memory usage but still has a long execution time. The third is optimized for low memory usage and fast execution.

different approach by leveraging LLMs to automatically generate new coding problems inspired by random code snippets from open-source repositories. In contrast, Octopack (Muennighoff et al., 2024) focused on collecting and filtering high-quality Git commit messages that resemble natural language instructions. While these existing methods primarily emphasize generating correct code, EFFICODER explores the use of fine-tuning to improve code efficiency. Our method is orthogonal to existing synthetic techniques, offering the potential for combination to further enhance the coding capabilities of LLMs.

## 3. EFFICODER: Fine-Tuning For Efficiency

### 3.1. Preliminary Study

We begin by investigating how the efficiency of training data influences the efficiency of code generated by LLMs. Following prior works (Huang et al., 2024b), we evaluate code efficiency using three metrics: Execution Time (ET), Max Memory Usage (MU), and Total Memory Usage (TMU). We hypothesize that training LLMs on efficient code will lead to the generation of more efficient code. To test this hypothesis, we synthesized multiple training datasets with varying levels of efficiency and used them to train different LLMs. The efficiency of the generated code was then measured in a controlled environment using ET, MU, and TMU. The training datasets included both efficient and inefficient code samples to ensure a comprehensive range of

efficiencies. The results, presented in Figure 3, reveal strong positive correlations between the efficiency of the training data and the efficiency of the generated code. Specifically, the correlation for ET is 0.972, for MU it is 0.950, and for TMU it is 0.986. These high correlation coefficients indicate that as the efficiency of the training data increases, so does the efficiency of the generated code. This study demonstrates that training LLMs on efficient code significantly enhances the efficiency of the generated code. The strong correlations across all three metrics support the hypothesis that the efficiency of the training data is a critical factor in improving the performance of LLM-generated code. These findings inspire further exploration of specific techniques for optimizing training datasets to maximize code efficiency.

### 3.2. Dataset Construction

**Curation Process** Figure 1 illustrates an overview of the process for constructing the EFFIINSTRUCT dataset for fine-tuning. The first step involves collecting candidate code generation tasks from nine open-source datasets available on the HuggingFace platform[1]. For each task, we aim to construct a more efficient solution compared to the initial solutions provided by the open-source datasets. Our approach shares similarities with existing works (Du et al., 2024; Huang et al., 2024a), where researchers execute LLM-generated code locally and analyze the execution time and

---

[1] https://huggingface.co/docs/datasets

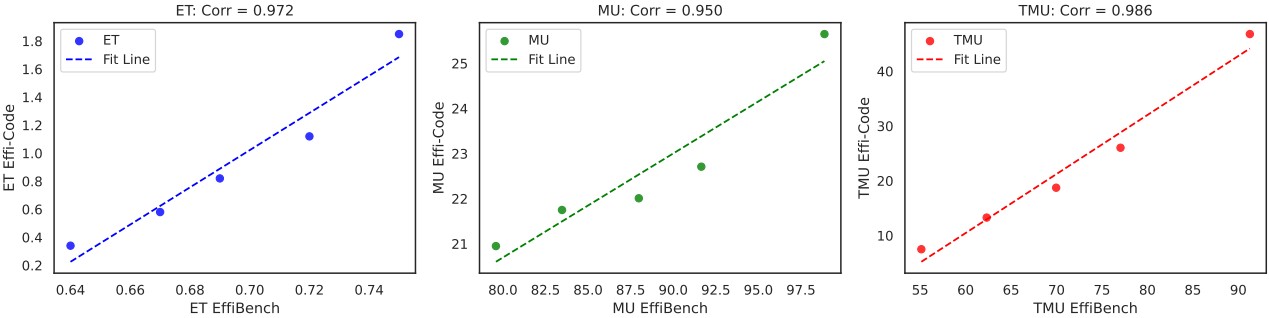

*Figure 3.* Correlation of the efficiency of the automatically generated code and the LLM code train set.

memory usage. However, our construction pipeline differs in that it utilizes multiple LLMs (e.g., DeepSeek-Coder and GPT-4o) to generate multiple candidate code solutions for each task in our candidate task set. We then directly calculate the execution time and memory usage for each generated code solution by executing them in local environments. The code with the lowest execution time and memory usage is selected as the final code for each task. For example, as shown in Figure 2, we directly select the code on the right as the final code.

**Data Sources** We collect the candidate tasks from the open-source code LLM training sets, which include SelfCodeAlign (SelfCodeAlign; Wei et al. 2024a), CodeFeedback-Filtered-Instruction (CodeFeed; MAP 2023), Tested-143k-Python-Alpaca (Alpaca; Vezora 2023), Glaive-Code-Assistant (Glaive; Computer 2023), Magicoder-Evol-Instruct-110K (Evol-Ins; UIUC 2023a), Dolphin-Coder (Dolphin; Computations 2023), Magicoder-OSS-Instruct-75K (Oss-Ins; UIUC 2023b), Self-OSS-Instruct-SC2-Exec-Filter-50K (Self-Oss; BigCode 2023), and Apps (Hendrycks et al., 2021). To collect the candidate tasks, all Python, C++, Java, Rust, and Go functions are extracted from the aforementioned open-source datasets. Following the filtering instructions of SelfCodeAlign (Wei et al., 2024a), a series of filtering rules are applied to ensure the code quality of the candidate tasks. After applying the filtering process, a total of 65k tasks were collected from an initial pool of about 790k candidate tasks[2].

### 3.3. Dataset Statistics

As shown in Table 1, coding problems in EFFIINSTRUCT have been collected from nine datasets, resulting in a total of 65,710 tasks across five programming languages: Python, C++, Java, Rust, and Go. The dataset encompasses a di-

verse range of coding challenges, ensuring a comprehensive coverage of various programming concepts and problem-solving techniques. Python has the highest representation in EFFIINSTRUCT, with 33,489 tasks sourced from all nine datasets. This extensive collection of Python tasks allows for effective fine-tuning of LLMs to generate efficient and optimized Python code. C++ and Java also have significant contributions, with 11,547 and 14,726 tasks, respectively. These tasks are primarily sourced from CodeFeed, Glaive, Evol-Ins, Dolphin, and Oss-Ins datasets, providing a robust foundation for fine-tuning LLMs in these popular programming languages. Rust and Go, although having relatively fewer tasks compared to Python, C++, and Java, still have a substantial presence in EFFIINSTRUCT. With 4,270 Rust tasks and 1,678 Go tasks, the dataset enables fine-tuning of LLMs to generate efficient code in these modern and rapidly growing programming languages.

Figure 4 illustrates the efficiency distribution of the dataset for three key metrics: execution time, memory usage, and max memory peak, which compares the distribution of these metrics for both inefficient (canonical solutions provided by the nine datasets) and efficient solutions in the EFFIINSTRUCT. For execution time, the inefficient solutions have a mean value of 1.14s, while the efficient solutions have a significantly lower mean of 0.31s, which indicates that the optimization process has successfully reduced the execution time of the code, resulting in more efficient solutions. Similarly, the memory usage and max memory peak also show a notable difference between inefficient and efficient solutions. For example, inefficient solutions have a mean memory usage of 26.50 MBs, while efficient solutions have a much lower mean of 6.03 MBs.

The efficiency distribution visualization highlights the effectiveness of the optimization process in creating more efficient solutions across all three metrics. By carefully curating tasks through the multi-step cleaning process and applying SOAP optimization, we have created a dataset that is valuable for training models to generate efficient

---
[2]Analysis shows no exact duplicates between training and evaluation sets, with only 0.20% of evaluation samples having minimal vocabulary overlap (5-10%).

*Table 1.* Distribution of tasks in the constructed EFFIINSTRUCT for different programming languages.

| Dataset | APPS | Alpaca | CodeFeed | Glaive | Evol-Ins | Dolphin | Oss-Ins | Self-Oss | SelfCodeAlign | Total |
|---|---|---|---|---|---|---|---|---|---|---|
| Python | 1001 | 2920 | 1387 | 32 | 1250 | 1958 | 76 | 827 | 24038 | 33489 |
| CPP | - | 3 | 1257 | 2675 | 3439 | 1186 | 2985 | 2 | - | 11547 |
| Java | - | 1 | 2082 | 3278 | 4692 | 1746 | 2927 | - | - | 14726 |
| Rust | - | - | 26 | 187 | 467 | 500 | 3090 | - | - | 4270 |
| Go | - | 1 | 47 | 277 | 776 | 549 | 28 | - | - | 1678 |

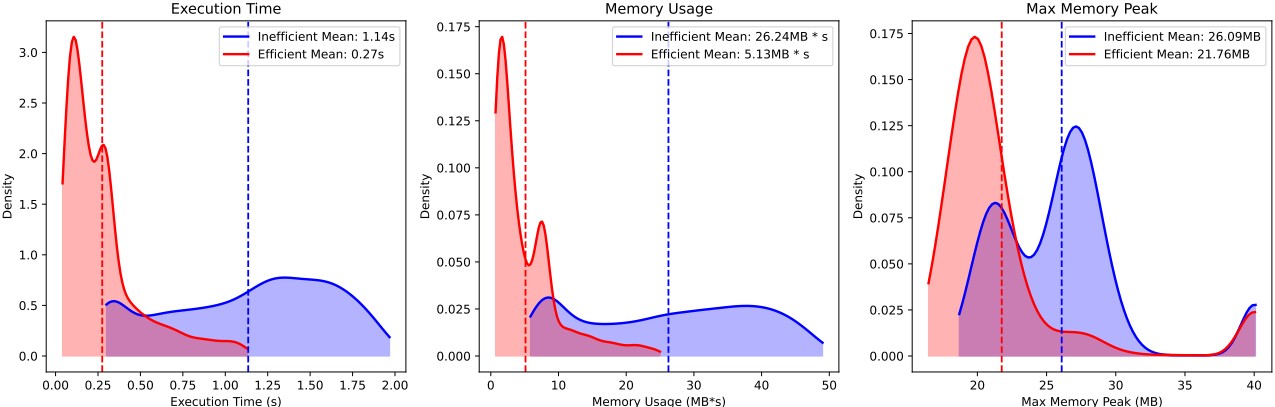

*Figure 4.* Efficiency distribution of the Python subset collected from Hugging Face. The figure shows the distribution of execution time, memory usage, and max memory peak for both inefficient (task-provided solution) and efficient solutions in the EFFIINSTRUCT. The inefficient solutions have higher overheads for all three metrics than the efficient ones.

code. EFFIINSTRUCT provides a diverse range of optimized coding problems, enabling researchers and practitioners to advance the field of code optimization using LLMs.

## 4. Experiment

**Datasets and Models** We evaluate the efficiency and correctness of LLM-generated code on five code generation benchmarks, i.e., EffiBench (Huang et al., 2024b), EvalPlus (HumanEvalPlus and MBPPPlus) (Liu et al., 2024a), DS-1000 (Lai et al., 2023), EvoEval (Xia et al., 2024), and HumanEval-X (Zheng et al., 2023). We finetune eight open-source LLMs with EFFIINSTRUCT, including CodeLlama-7b-bf, DeepSeek-Coder-6.7B base and instruct model (DeepSeekAI, 2023), Qwen2.5-Coder-7B base and instruct model (Hui et al., 2024), and Qwen2.5-Coder (1.5B, 3B, and 14B).

**Fine-tuning Setup** We use Llama-factory (Zheng et al., 2024) to fine-tune LLMs with fully supervised fine-tuning with the same setup and train the models using EFFIIN-STRUCT. The maximum sequence length is set to 2048 tokens. We use a batch size of 128 and set the learning rate to 5e-6 with a cosine learning rate scheduler and a warmup ratio of 0.03. We fine-tune all LLMs for 4 epochs under the bf16 data type.

### 4.1. Evaluation of Python Code

To comprehensively demonstrate the efficiency and correctness of automatically generated code by LLMs with the SFT of EFFIINSTRUCT, we first provide the evaluation of the LLMs in generating Python code, where LLMs are asked to create Python code based on natural language or function signatures with docstrings.

**EffiBench** is a benchmark used to measure the efficiency and correctness of LLM-generated code in LeetCode tasks. To ensure that the efficiency of LLM-generated code can be measured, the authors construct 100 tests for each task to provide enough testing time. As shown in Table 2, we can observe that for all LLMs, the efficiency of the LLM-generated code has been improved after fine-tuning with EFFIINSTRUCT. For example, the average execution time (ET) for the correct code generated by Qwen2.5-Coder-7B-Instruct and its EFFIINSTRUCT fine-tuned version decreases from 0.31 (s) to 0.16 (s), a reduction of 48.4%. Similarly, the total memory usage (TMU) of LLM-generated code also shows significant decreases. For instance, the TMU of DeepSeek-Coder-6.7B-Instruct decreases from 30.05 (Mb*s) to 9.48 (Mb*s), a larger reduction than the decrease in average execution time for the correct code generated by the same model, which only reduces by 35.3% from

*Table 2.* Code efficiency and pass@1 of LLMs trained with EFFIINSTRUCT on EffiBench using greedy decoding. The percentage in the brackets indicates the extent of the reduction for each respective item. `Overlap` means the percentage of correct tasks addressed by both EFFIINSTRUCT finetuned LLM and original LLM in total tasks of the dataset.

| Model | ET (s) ↓ | NET ↓ | MU (Mb) ↓ | NMU ↓ | TMU (Mb*s) ↓ | NTMU ↓ | Overlap (%) ↑ | Pass@1 (%) ↑ |
|---|---|---|---|---|---|---|---|---|
| CodeLlama-7b-hf | 0.24 | 1.48 | 141.91 | 4.78 | 149.23 | 73.82 | 8.2 | 15.0 |
| + EFFIINSTRUCT | 0.20 (16.7%) | 1.27 (14.2%) | 80.86 (43.0%) | 2.26 (52.7%) | 87.09 (41.6%) | 42.66 (42.2%) | 8.2 | 17.6 |
| deepseek-coder-6.7b-base | 0.37 | 2.62 | 36.94 | 1.04 | 17.26 | 2.74 | 47.1 | 54.4 |
| + EFFIINSTRUCT | 0.21 (43.2%) | 1.42 (45.8%) | 37.11 (-0.5%) | 1.04 (0.0%) | 11.97 (30.6%) | 2.10 (23.4%) | 47.1 | 59.3 |
| deepseek-coder-6.7b-instruct | 0.34 | 2.56 | 47.26 | 1.45 | 30.05 | 9.97 | 36.0 | 44.4 |
| + EFFIINSTRUCT | 0.22 (35.3%) | 1.71 (33.2%) | 36.31 (23.2%) | 1.00 (31.0%) | 9.48 (68.5%) | 2.11 (78.8%) | 36.0 | 51.7 |
| Qwen2.5-Coder-7B-Instruct | 0.31 | 2.35 | 31.66 | 1.00 | 11.00 | 2.15 | 37.2 | 44.8 |
| + EFFIINSTRUCT | 0.16 (48.4%) | 1.12 (52.3%) | 31.67 (-0.0%) | 1.00 (0.0%) | 8.28 (24.7%) | 1.18 (45.1%) | 37.2 | 57.7 |
| Qwen2.5-Coder-1.5B | 0.40 | 2.95 | 35.34 | 1.03 | 15.68 | 3.51 | 27.1 | 39.6 |
| + EFFIINSTRUCT | 0.22 (45.0%) | 1.60 (45.8%) | 34.58 (2.2%) | 1.00 (2.9%) | 9.09 (42.0%) | 1.98 (43.6%) | 27.1 | 41.7 |
| Qwen2.5-Coder-3B | 0.43 | 2.73 | 48.73 | 1.00 | 33.88 | 2.59 | 16.9 | 31.2 |
| + EFFIINSTRUCT | 0.23 (46.5%) | 1.60 (41.4%) | 49.18 (-0.9%) | 1.00 (0.0%) | 18.31 (46.0%) | 1.98 (23.6%) | 16.9 | 34.2 |
| Qwen2.5-Coder-7B | 0.26 | 1.81 | 38.88 | 1.01 | 18.63 | 3.01 | 41.4 | 50.1 |
| + EFFIINSTRUCT | 0.17 (34.6%) | 1.23 (32.0%) | 38.61 (0.7%) | 1.00 (1.0%) | 10.82 (41.9%) | 1.32 (56.1%) | 41.4 | 57.3 |
| Qwen2.5-Coder-14B | 0.36 | 2.73 | 32.41 | 1.00 | 12.59 | 2.57 | 50.1 | 57.5 |
| + EFFIINSTRUCT | 0.15 (58.3%) | 1.14 (58.2%) | 32.41 (0.0%) | 1.00 (0.0%) | 6.80 (46.0%) | 1.23 (52.1%) | 50.1 | 63.6 |

*Table 3.* Code efficiency and pass@1 of LLMs trained with EFFIINSTRUCT on HumanEvalPlus and MBPPPlus.

| Model | ET (s) ↓ | NET ↓ | MU (Mb) ↓ | NMU ↓ | TMU (Mb*s) ↓ | NTMU ↓ | Overlap (%) ↑ | Pass@1 (%) ↑ |
|---|---|---|---|---|---|---|---|---|
| HumanEvalPlus | | | | | | | | |
| deepseek-coder-6.7b-base | 0.43 | 1.04 | 67.45 | 1.00 | 28.23 | 1.02 | 7.3 | 7.3 |
| + EFFIINSTRUCT | 0.42 (2.3%) | 1.00 (3.8%) | 67.37 (0.1%) | 1.00 (0.0%) | 27.90 (1.2%) | 1.02 (0.0%) | 7.3 | 64.6 |
| deepseek-coder-6.7b-instruct | 0.54 | 2.27 | 61.64 | 0.98 | 20.18 | 2.30 | 42.1 | 47.6 |
| + EFFIINSTRUCT | 0.37 (31.5%) | 1.45 (36.1%) | 61.58 (0.1%) | 0.98 (0.0%) | 16.48 (18.3%) | 1.78 (22.6%) | 42.1 | 72.6 |
| Qwen2.5-Coder-7B | 0.35 | 1.23 | 61.77 | 0.98 | 15.25 | 1.39 | 36.6 | 40.2 |
| + EFFIINSTRUCT | 0.29 (17.1%) | 0.96 (22.0%) | 61.70 (0.1%) | 0.98 (0.0%) | 12.18 (20.1%) | 0.96 (30.9%) | 36.6 | 78.7 |
| Qwen2.5-Coder-7B-Instruct | 0.52 | 2.05 | 63.38 | 0.99 | 20.17 | 1.96 | 67.7 | 76.2 |
| + EFFIINSTRUCT | 0.32 (38.5%) | 1.08 (47.3%) | 63.35 (0.0%) | 0.99 (0.0%) | 15.15 (24.9%) | 1.16 (40.8%) | 67.7 | 78.0 |
| MBPPPlus | | | | | | | | |
| deepseek-coder-6.7b-base | 0.49 | 1.64 | 58.90 | 1.00 | 17.27 | 1.62 | 55.3 | 63.2 |
| + EFFIINSTRUCT | 0.31 (36.7%) | 0.97 (40.9%) | 58.99 (-0.2%) | 1.00 (0.0%) | 10.27 (40.5%) | 0.96 (40.7%) | 55.3 | 65.9 |
| deepseek-coder-6.7b-instruct | 0.43 | 1.65 | 59.03 | 1.00 | 14.39 | 1.65 | 59.0 | 65.3 |
| + EFFIINSTRUCT | 0.31 (27.9%) | 1.01 (38.8%) | 58.97 (0.1%) | 1.00 (0.0%) | 10.35 (28.1%) | 1.02 (38.2%) | 59.0 | 67.5 |
| Qwen2.5-Coder-7B | 0.48 | 1.70 | 58.89 | 0.99 | 17.00 | 1.79 | 59.5 | 60.1 |
| + EFFIINSTRUCT | 0.31 (35.4%) | 0.96 (43.5%) | 58.98 (-0.2%) | 0.99 (0.0%) | 10.33 (39.2%) | 0.94 (47.5%) | 59.5 | 63.2 |
| Qwen2.5-Coder-7B-Instruct | 0.46 | 1.68 | 64.90 | 1.00 | 23.99 | 1.66 | 63.2 | 68.0 |
| + EFFIINSTRUCT | 0.30 (34.8%) | 0.96 (42.9%) | 68.31 (-5.3%) | 1.00 (0.0%) | 16.91 (32.4%) | 0.95 (42.8%) | 63.2 | 70.6 |

0.34 (s) to 0.22 (s). This indicates that during code generation, both the execution time and memory usage of the fine-tuned-LLM-generated code have been improved compared to the code generated by the original models. Furthermore, the memory usage/peak (MU) of DeepSeek-Coder-6.7B-Instruct generated code decreases from 47.26 (Mb) to 36.31 (Mb), a reduction of 23.2%, ensuring that the LLM-generated code can be deployed in memory-constrained scenarios such as embedded systems or edge devices.

Interestingly, we observe that compared to the MU, ET is more widely optimized across all models. This suggests that EFFIINSTRUCT fine-tuning has a more significant impact on reducing the execution time of the generated code than on reducing its memory footprint. Nevertheless, the improvements in execution time and memory usage demonstrate the effectiveness of EFFIINSTRUCT in enhancing the efficiency of LLM-generated code.

**HumanEvalPlus and MBPPPlus** As shown in Table 3, we observe that almost all LLMs achieve better efficiency and

higher correctness after being fine-tuned with . Take HumanEvalPlus as an example, the average execution time (ET) for correct code generated by Qwen2.5-Coder-7B-Instruct and its fine-tuned version decreases from 0.52 (s) to 0.32 (s), a reduction of 38.5%. These improvements demonstrate the effectiveness of in optimizing the efficiency of LLM-generated code. Moreover, the pass@1 of Qwen2.5-Coder-7B increases from 40.2% to 78.7%, an improvement of 38.5% after fine-tuning with . This indicates that the fine-tuned models not only generate more efficient code but also produce correct code more frequently. Similar to the results in HumanEvalPlus, the efficiency and correctness of fine-tuned LLMs also improve in the MBPPPlus dataset. For instance, the average execution time for correct code generated by deepseek-coder-6.7b-base decreases by 36.7%, and the total memory usage (TMU) decreases by 40.5% after fine-tuning.

**EvoEval** includes 828 programming problems created by prompting GPT-4 to evolve original HumanEval tasks across

*Table 4.* Code efficiency and pass@1 of Qwen2.5-Coder-7B-Instruct and deepseek-coder-6.7b-instruct fine-tuned using SFT with the EFFIINSTRUCT for the EvoEval dataset.

| Model | ET (s) ↓ | NET ↓ | MU (Mb) ↓ | NMU ↓ | TMU (Mb*s) ↓ | NTMU ↓ | Overlap (%) ↑ | Pass@1 (%) ↑ |
|---|---|---|---|---|---|---|---|---|
| EvoEval_tool_use | | | | | | | | |
| Qwen2.5-Coder-7B-Instruct | 0.35 | 1.80 | 59.11 | 1.00 | 11.22 | 1.74 | 33.0 | 43.0 |
| + EFFIINSTRUCT | 0.20 (42.9%) | 0.98 (45.6%) | 59.08 (0.1%) | 1.00 (0.0%) | 6.55 (41.6%) | 0.99 (43.1%) | 33.0 | 53.0 |
| deepseek-coder-6.7b-instruct | 0.41 | 1.83 | 74.88 | 1.00 | 31.51 | 1.74 | 44.0 | 54.0 |
| + EFFIINSTRUCT | 0.24 (41.5%) | 0.99 (45.9%) | 74.77 (0.1%) | 1.00 (0.0%) | 26.50 (15.9%) | 1.00 (42.5%) | 44.0 | 55.0 |
| EvoEval_subtle | | | | | | | | |
| Qwen2.5-Coder-7B-Instruct | 0.40 | 1.49 | 70.46 | 1.00 | 29.15 | 1.46 | 48.0 | 55.0 |
| + EFFIINSTRUCT | 0.33 (17.5%) | 1.11 (25.5%) | 70.47 (-0.0%) | 1.00 (0.0%) | 28.30 (2.9%) | 1.17 (19.9%) | 48.0 | 72.0 |
| deepseek-coder-6.7b-instruct | 0.45 | 1.97 | 59.06 | 0.99 | 17.01 | 2.00 | 50.0 | 56.0 |
| + EFFIINSTRUCT | 0.30 (33.3%) | 1.32 (33.0%) | 58.93 (0.2%) | 0.99 (0.0%) | 11.19 (34.2%) | 1.31 (34.5%) | 50.0 | 69.0 |
| EvoEval_creative | | | | | | | | |
| Qwen2.5-Coder-7B-Instruct | 0.51 | 2.16 | 62.71 | 1.00 | 24.21 | 2.39 | 32.0 | 44.0 |
| + EFFIINSTRUCT | 0.37 (27.5%) | 1.45 (32.9%) | 62.69 (0.0%) | 1.00 (0.0%) | 21.10 (12.8%) | 1.82 (23.8%) | 32.0 | 44.0 |
| deepseek-coder-6.7b-instruct | 0.42 | 1.71 | 61.88 | 1.00 | 15.91 | 1.62 | 27.0 | 31.0 |
| + EFFIINSTRUCT | 0.26 (38.1%) | 0.99 (42.1%) | 61.75 (0.2%) | 1.00 (0.0%) | 10.86 (31.7%) | 0.98 (39.5%) | 27.0 | 41.0 |

*Table 5.* Code efficiency and pass@1 of Qwen2.5-Coder-7B-Instruct and deepseek-coder-6.7b-instruct fine-tuned using SFT with the EFFIINSTRUCT for the DS-1000 dataset.

| Model | ET (s) ↓ | NET ↓ | MU (Mb) ↓ | NMU ↓ | TMU (Mb*s) ↓ | NTMU ↓ | Overlap (%) ↑ | Pass@1 (%) ↑ |
|---|---|---|---|---|---|---|---|---|
| Qwen2.5-Coder-7B-Instruct | 1.2668 | 1.0102 | 447.7101 | 1.0148 | 6.831366 | 1.0175 | 9.30 | 17.00 |
| + EFFIINSTRUCT | 1.2593 (0.8%) | 1.0041 (1.0%) | 447.0711 (0.1%) | 1.0134 (0.0%) | 6.7694 (0.9%) | 1.0081 (1.0%) | 9.30 | 35.70 |
| deepseek-coder-6.7b-instruct | 1.3154 | 1.0536 | 450.2792 | 1.0206 | 7.204722 | 1.0731 | 29.10 | 37.50 |
| + EFFIINSTRUCT | 1.2587 (4.5%) | 1.0082 (3.8%) | 441.7553 (1.9%) | 1.0013 (2.0%) | 6.8022 (5.6%) | 1.0132 (5.6%) | 29.10 | 37.50 |

5 semantic-altering and 2 semantic-preserving benchmarks, each of which has 100 problems. We conduct experiments on the Tool_Use, Subtle, and Creative benchmarks to evaluate the performance of Qwen2.5-Coder-7B-Instruct and deepseek-coder-6.7b-instruct fine-tuned with . As shown in Table 4, both models demonstrate significant improvements in code efficiency after fine-tuning with . For the Tool_Use benchmark, the average execution time (ET) for correct code generated by Qwen2.5-Coder-7B-Instruct decreases by 42.9%, and the total memory usage (TMU) decreases by 41.6% after fine-tuning. Similarly, deepseek-coder-6.7b-instruct achieves a 41.5% reduction in ET and a 15.9% reduction in TMU. In the Subtle benchmark, after fine-tuning, Qwen2.5-Coder-7B-Instruct and deepseek-coder-6.7b-instruct exhibit 17.5% and 33.3% reductions in ET, respectively, after fine-tuning. The normalized total memory usage (NTMU) also decreases by 19.9% and 34.5% for the two models. Moreover, the pass@1 rate improves significantly, with Qwen2.5-Coder-7B-Instruct increasing from 55.0% to 72.0% and deepseek-coder-6.7b-instruct increasing from 56.0% to 69.0%. For the Creative benchmark, Qwen2.5-coder-7B-instruct and deepseek-coder-6.7b-instruct achieve 27.5% and 38.1% reductions in ET, respectively, after fine-tuning. The NTMU also decreases by 23.8% and 39.5% for the two models. The pass@1 rate remains the same for Qwen2.5-Coder-7B-Instruct at 44.0% but improves from 31.0% to 41.0% for deepseek-coder-6.7b-instruct.

## 4.2. Data Science Programming

DS-1000 is a data science benchmark consisting of 1000 realistic challenges across 7 popular Python data science libraries. We evaluate the efficiency and pass@1 of LLM-generated code for the DS-1000 tasks using Qwen2.5-Coder-7B-Instruct and deepseek-coder-6.7b-instruct, both in their original form and fine-tuned with . As shown in Table 5, fine-tuning with leads to modest improvements in code efficiency for both models. For Qwen2.5-Coder-7B-Instruct, the average execution time (ET) decreases by 0.8%, and the total memory usage (TMU) decreases by 0.9% after fine-tuning, while the pass@1 rate improves significantly from 17.00% to 35.70%. For deepseek-coder-6.7b-instruct, fine-tuning results in a 4.5% reduction in ET and a 5.6% reduction in TMU, with the memory usage (MU) and normalized memory usage (NMU) also decreasing by 1.9% and 2.0%, respectively, although the pass@1 rate remains the same at 37.50%. While the improvements in code efficiency for the DS-1000 benchmark are less pronounced compared to other benchmarks, the results still demonstrate that fine-tuning with can enhance the efficiency of LLM-generated code for data science tasks.

## 4.3. Different Programming Language

In addition to evaluating the efficiency and pass@1 of LLM-generated code for Python tasks, we also conduct experiments on the HumanEval-X dataset, where we focus on the C++ and Java subsets, to measure the performance of

*Table 6.* Code efficiency and pass@1 of Qwen2.5-Coder-7B-Instruct and deepseek-coder-6.7b-instruct fine-tuned using SFT with the EFFIINSTRUCT for the HumanEval-X (CPP and Java) dataset.

| Model | ET (s) ↓ | NET ↓ | MU (Mb) ↓ | NMU ↓ | TMU (Mb*s) ↓ | NTMU ↓ | Overlap (%) ↑ | Pass@1 (%) ↑ |
|---|---|---|---|---|---|---|---|---|
| CPP | | | | | | | | |
| Qwen2.5-Coder-7B-Instruct | 0.0015 | 1.3973 | 1.5223 | 1.9489 | 0.000013 | 5.195024 | 4.27 | 7.32 |
| + EFFIINSTRUCT | 0.0009 (37.1%) | 1.0138 (27.4%) | 0.8739 (42.6%) | 1.0585 (45.7%) | 0.000006 (53.8%) | 2.784936 (46.4%) | 4.27 | 48.17 |
| deepseek-coder-6.7b-instruct | 0.0015 | 1.2417 | 1.2991 | 1.0731 | 0.000010 | 2.078790 | 4.27 | 14.63 |
| + EFFIINSTRUCT | 0.0008 (45.2%) | 0.5312 (57.2%) | 0.6496 (50.0%) | 0.4289 (60.0%) | 0.000006 (40.0%) | 0.543047 (73.9%) | 4.27 | 40.24 |
| Java | | | | | | | | |
| Qwen2.5-Coder-7B-Instruct | - | - | - | - | - | - | - | - |
| + EFFIINSTRUCT | 0.0082 | 0.2037 | 8.5926 | 0.1918 | 0.002996 | 0.1859 | - | 57.93 |
| deepseek-coder-6.7b-instruct | 0.0076 | 0.1921 | 8.1062 | 0.1789 | 0.002731 | 0.180034 | 6.71 | 14.63 |
| + EFFIINSTRUCT | 0.0048 (36.5%) | 0.1231 (35.9%) | 4.0142 (50.5%) | 0.0908 (49.2%) | 0.001900 (30.4%) | 0.118972 (33.9%) | 6.71 | 57.93 |

*Table 7.* Efficiency comparison of different methods on the HumanEvalPlus dataset. We use the fine-tuned CodeLlama-7b-hf by PIE and Mercury as the baselines to measure the improvement of EFFIINSTRUCT fine-tuned version.

| Method | ET | NET | MU | NMU | TMU | NTMU | overlapped | pass@1 |
|---|---|---|---|---|---|---|---|---|
| CodeLlama-7b-hf | 0.39 | 1.94 | 61.68 | 1.00 | 12.78 | 1.83 | 9.8 | 12.2 |
| + PIE (All) | 0.30 (23.1%) | 1.47 (24.2%) | 61.39 (0.5%) | 1.00 (0.0%) | 11.28 (11.7%) | 1.68 (8.2%) | 9.8 | 19.5 |
| + EFFIINSTRUCT | 0.21 (46.2%) | 1.03 (46.9%) | 61.33 (0.6%) | 1.00 (0.0%) | 7.17 (43.9%) | 1.04 (43.2%) | 9.8 | 31.1 |
| CodeLlama-7b-hf | 0.39 | 1.94 | 61.69 | 1.00 | 12.78 | 1.83 | 4.3 | 12.2 |
| + Mercury | 0.31 (20.5%) | 1.51 (22.2%) | 61.94 (-0.4%) | 1.00 (0.0%) | 10.24 (19.9%) | 1.47 (19.7%) | 4.3 | 9.1 |
| + EFFIINSTRUCT | 0.21 (46.2%) | 1.01 (47.9%) | 61.73 (-0.1%) | 1.00 (0.0%) | 6.95 (45.6%) | 0.98 (46.4%) | 4.3 | 31.1 |

fine-tuned LLMs in a different-language setting. As shown in Table 6, both Qwen2.5-Coder-7B-Instruct and deepseek-coder-6.7b-instruct demonstrate significant improvements in code efficiency and pass@1 rates after fine-tuning with for C++ tasks. Qwen2.5-Coder-7B-Instruct achieves a 37.1% reduction in average execution time (ET), a 42.6% reduction in memory usage (MU), and a 53.8% reduction in total memory usage (TMU), with the pass@1 rate improving from 7.32% to 48.17%. Similarly, deepseek-coder-6.7b-instruct exhibits a 45.2% reduction in ET, a 50.0% reduction in MU, and a 40.0% reduction in TMU, with the pass@1 rate increasing from 14.63% to 40.24%. For Java tasks, deepseek-coder-6.7b-instruct achieves a 36.5% reduction in ET, a 50.5% reduction in MU, and a 30.4% reduction in TMU after fine-tuning, with the pass@1 rate improving from 14.63% to 57.93%, matching the performance of the fine-tuned Qwen2.5-Coder-7B-Instruct. These results demonstrate that fine-tuning with can significantly enhance the efficiency and correctness of LLM-generated code across multiple programming languages.

### 4.4. Comparison with Baselines

To further demonstrate the efficiency of the code generated by fine-tuned LLMs, we compare the performance of CodeLlama-7b-hf fine-tuned with two baselines: PIE (Shypula et al., 2024) and Mercury (Du et al., 2024). Similar with EFFICODER, PIE and Mercury are fine-tuned to improve the efficiency of LLM-generated code. The evaluation results on the HumanEvalPlus dataset are presented in Table 7. We can observe that for the correct tasks addressed by both PIE and

PIE required 0.30 (s) on average to address each task, which is a 23.1% reduction in average execution time compared to the original CodeLlama-7b-hf generated code. However, the fine-tuned CodeLlama-7b-hf reduces the average execution time by 46.2%, requiring only 0.21 (s) on average to address each correct task. Moreover, the fine-tuned model achieves a 46.9% reduction in NET, a 43.9% reduction in TMU, and a 43.2% reduction in NTMU compared to PIE. The pass@1 also improves from 19.5% for PIE to 31.1% for the fine-tuned model. Similarly, when compared to Mercury, the fine-tuned CodeLlama-7b-hf demonstrates a 46.2% reduction in average execution time, a 47.9% reduction in NET, a 45.6% reduction in TMU, and a 46.4% reduction in NTMU. The pass@1 rate improves significantly from 9.1% for Mercury to 31.1% for the fine-tuned model, with both methods having an overlapped percentage of 4.3%. The substantial improvements in efficiency and correctness achieved by the fine-tuned model demonstrate the effectiveness of this approach in optimizing LLM-generated code for practical applications.

EFFICODER's exceptional performance can be attributed to several key factors. First, EFFICODER was designed with multilingual programming support in mind, covering a diverse range of programming languages in its training process. This comprehensive language coverage enables the model to perform consistently well across various language-specific test sets, demonstrating its versatility and broad applicability in real-world programming scenarios. Second, EFFICODER's superior efficiency stems from our meticulous data processing method. During the data preparation

phase, we implemented rigorous filtering operations that specifically retained only tasks demonstrating measurable improvements in code efficiency (as illustrated in our motivation figure). This efficiency-focused data curation strategy directly aligns with our core objective of optimizing computational performance. Third, the scale of our training dataset significantly outperforms previous approaches, with over 70,000 training examples compared to Mercury's mere 1,800 samples. This substantial difference in training data volume allows our fine-tuned language models to develop more robust patterns for generating efficient code across diverse contexts. The combination of these factors, multilingual support, efficiency-oriented data filtering, and large-scale training data, collectively contribute to EFFICODER's ability to consistently outperform baseline methods in both code correctness and execution efficiency metrics.

## 5. Conclusion

In this paper, our research addresses a critical gap in the efficiency of code generated by LLMs by introducing the EFFIINSTRUCT dataset, designed to enhance both the correctness and execution efficiency of LLM-generated code via fine-tuning. Through meticulous aggregation, preprocessing, and iterative optimization, we provide a robust resource that significantly boosts the performance of open-source LLMs like DeepSeek-Coder and Qwen. Our experiments reveal substantial improvements, with notable increases in pass rates and decreases in execution time, underscoring the potential of EFFICODER to advance the state of code generation in resource-constrained environments. By open-sourcing our model weights, training data, and source code, we aim to foster further research and innovation in this vital area of AI development tools.

## Impact Statement

This paper presents work aimed at advancing the field of Machine Learning by improving the efficiency and correctness of code generated by LLMs. The societal benefits of this research include:

- Sustainability: By reducing computational resource consumption (e.g., energy and memory usage), our work aligns with global efforts to mitigate the environmental impact of large-scale computing. Efficient code generation could lower operational costs and energy demands in industries reliant on software development.

- Resource-Constrained Environments: Enhanced efficiency enables broader adoption of LLM-generated code in mobile, embedded, or edge computing systems, where energy and processing power are limited.

- Research Advancement: Open-sourcing our dataset

and models fosters transparency and accelerates research into sustainable AI systems, encouraging further innovations in code optimization.

Ethical Considerations:

- Data Provenance: The dataset is aggregated from publicly available open-source repositories on Hugging Face, ensuring compliance with licensing terms. However, future work should continue to prioritize responsible data curation practices.

- Bias and Generalization: While our framework supports multilingual adaptability, biases in the source code (e.g., language-specific optimizations or cultural coding norms) may inadvertently propagate. Mitigating such biases requires careful dataset design and validation.

- Developer Dependency: Widespread adoption of optimized LLM-generated code could influence coding practices. Ensuring human developers retain critical problem-solving skills remains important.

Overall, this work aims to address a critical gap in LLM-generated code efficiency while maintaining correctness. We encourage future research to explore trade-offs between efficiency, maintainability, and fairness in automated code generation.

## Acknowledgment

The work is supported in part by National Key R&D Program of China (2022ZD0160201), HK RGC RIF (R7030-22), HK RGC GRF (ref No.: 17208223 & 17204424), a Huawei flagship research grant in 2023, SupernetAI, and the HKU-CAS Joint Laboratory for Intelligent System Software. This work is also supported by the National Research Foundation, Singapore under its AI Singapore Programme (AISG Award No: AISG2-PhD-2021-08-007) and ITEA Genius and ITEA GreenCode projects, funded by InnovateUK.

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

# A. Appendix

## A.1. Prompt Template

> Please continue to complete the function. You are not allowed to modify the given code and do the completion only. Please return all completed functions in a code block. Here is the given code to complete:
> ```python
> {{Prompt}}
> ```

## A.2. Efficiency Metrics

**Execution Time (ET)**  Execution time (ET) measures the average time taken for code execution. Mathematically, ET is defined as:

$$ET = \frac{1}{N} \sum^{N} T_{\text{code}}$$

where $ET$ is the execution time metric, $T_{\text{code}}$ is the execution time of the code (with all the test cases), and $N$ is the number of codes generated by code generation models used for evaluation.

**Normalized Execution Time (NET)**  Normalized Execution Time (NET) measures the execution time required by generated code relative to that of a canonical solution. We define NET as:

$$NET = \frac{1}{N} \sum^{N} \frac{T_{\text{code}}}{T_{\text{canonical}}}$$

where $T_{\text{code}}$ is the execution time of the generated code and $T_{\text{canonical}}$ is the execution time of the canonical solution. A NET value greater than 1 indicates that the generated code is slower than the canonical solution, while a value less than 1 suggests the generated code is faster.

**Max Memory Usage (MU)**  Max Memory Usage (MU) measures the average max memory consumption during code execution. Mathematically, MU is defined as:

$$MU = \frac{1}{N} \sum^{N} M_{\text{code}}$$

where $MU$ is the memory usage metric, $M_{\text{code}}$ is the max memory consumption of the generated code among all the test cases, and $N$ is the number of code instances generated by code generation models used for evaluation. This metric is critical to assess the resource efficiency of generated code, particularly in environments with limited maximum memory capacity.

**Normalized Max Memory Usage (NMU)**  Normalized Max Memory Usage (NMU) quantifies how the max memory efficiency of the generated code compares to the canonical solution. We define NMU as:

$$NMU = \frac{1}{N} \sum^{N} \frac{M_{\text{code}}}{M_{\text{canonical}}}$$

where $NMU$ is the normalized max memory usage metric, $M_{\text{code}}$ is the max memory usage of the generated code, and $M_{\text{canonical}}$ is the max memory usage of the canonical solution. An NMU value less than 1 indicates that the generated code is more memory-efficient than the canonical solution, whereas a value greater than 1 suggests it is less efficient in terms of memory usage. This metric provides a relative measure of the memory optimization in the generated code in comparison to a standard baseline.

**Total Memory Usage (TMU)**  Total Memory Usage (TMU) assesses the efficiency of memory usage throughout the execution of code, taking into account both the magnitude and duration of memory utilization. To calculate TMU, first, monitor and record the memory usage at discrete time intervals during the execution, resulting in a memory usage profile

$M(t)$, where $t$ represents time. Then, compute the area under the curve of $M(t)$ over the total execution time, $T_{\text{total}}$, using numerical integration methods such as the trapezoidal rule:

$$TMU = \frac{1}{N} \sum^{N} \int_{0}^{T_{\text{total}}} M(t)\, dt$$

A lower TMU value indicates higher memory efficiency, reflecting an optimized balance between the amount of memory used and the duration of its usage.

**Normalized Total Memory Usage (NTMU)**  The Normalized Total Memory Usage (NTMU) offers a comparison of the dynamic memory efficiency between the generated code and the canonical solution. To determine NTMU, calculate the TMU for both the generated code and the canonical solution. Normalize the TMU of the generated code by dividing it by the TMU of the canonical solution:

$$NTMU = \frac{1}{N} \sum^{N} \frac{TMU_{\text{code}}}{TMU_{\text{canonical}}}$$

where $TMU_{\text{code}}$ is the TMU of the generated code and $TMU_{\text{canonical}}$ is the TMU of the canonical solution. An NTMU value less than 1 signifies that the generated code manages dynamic memory more efficiently compared to the canonical solution, while a value greater than 1 indicates less efficient management of dynamic memory. This metric provides insight into the relative use of dynamic memory of generated code compared to an established benchmark.

### A.3. Robustness of Overhead Results

The overhead results would be affected by the local environments, which causes that the results of Effi-Code fine-tuned LLMs may not able to represent the results of the efficiency profiling in different environments. To address this issue, we have conducted additional experiments and provided more robust evaluation results.

Firstly, we have evaluated the effectiveness of Effi-Code on seven different software-hardware setups, as shown in Rebuttal Table 2. The results demonstrate that Effi-Code fine-tuned LLMs achieve higher efficiency than the original LLMs across all setups. For example, in the environment of Python 3.11.10 - Intel(R) Xeon(R) Platinum 8336C CPU @ 2.30GHz, the average execution time decreases from 0.59s to 0.40s when using Effi-Code to fine-tune Qwen2.5-Coder-7B, reducing the average execution time by 32%.

Secondly, we clarify that for the same setup, where we evaluate the efficiency of LLM-generated code several times, the efficiency results are consistent. As shown in Paper Table 8, where we execute the LLM-generated code five times, the standard deviation of execution time (ET) is 0.00548 (s), indicating that the evaluation results are consistent and reliable for a given setup.

Finally, our evaluation setup follows the practices established in recent works on benchmarking the efficiency of automatically generated code, such as Mercury (Du et al., 2024), Effibench (Huang et al., 2024b), and SOAP (Huang et al., 2024a). By adhering to these benchmarks, we ensure that our evaluation is in line with the current standards in the field.

### A.4. Test case augmentation

Some of the candidate tasks we collected do not have test cases. To address this, we use GPT-3.5-turbo to construct test cases by feeding the task description and source code into the model and requiring it to generate test cases for our experiments. After that, we analyze whether each test case generated by GPT-3.5-turbo is correct and then filter out incorrect test cases and tasks that do not have the correct test cases. To determine the correctness of the test cases generated by GPT-3.5-turbo, we execute each test case individually with the initial solution for each task in our collected candidate tasks. We check whether any errors are raised during the execution of each test case with the initial solution. In other words, we verify if the test case passes the initial solution. We treat the test cases that pass the initial solution as correct. On the other hand, test cases that do not pass the initial solution are filtered out. By using the initial solution as a reference, we can effectively assess the correctness of the generated test cases and ensure that only valid test cases are retained for further analysis.

### A.5. Comparison with PIE strategies

In Table 7, we compared against the best-performing PIE model, which was the "All" strategy fine-tuned CodeLlama-7B in our evaluation. Now, we included comparisons with other PIE fine-tuning strategies in Table 9. EFFIINSTRUCT outperforms

| Setup | ET | NET | MU | NMU | TMU | NTMU |
|---|---|---|---|---|---|---|
| Python 3.11.10 - Intel(R) Xeon(R) Platinum 8336C CPU @ 2.30GHz | | | | | | |
| Qwen2.5-Coder-7B | 0.59 | 1.95 | 61.95 | 0.99 | 24.29 | 1.83 |
| + EFFIINSTRUCT | 0.40 | 1.01 | 61.96 | 0.99 | 18.74 | 1.02 |
| Python 3.11.10 - Intel(R) Xeon(R) Silver 4216 CPU @ 2.10GHz | | | | | | |
| Qwen2.5-Coder-7B | 0.28 | 1.63 | 36.15 | 1.00 | 20.01 | 1.88 |
| + EFFIINSTRUCT | 0.25 | 1.38 | 36.52 | 1.01 | 19.85 | 1.56 |
| Python 3.11.10 - Intel(R) Xeon(R) Silver 4116 CPU @ 2.10GHz | | | | | | |
| Qwen2.5-Coder-7B | 0.35 | 1.45 | 36.14 | 1.00 | 24.28 | 1.63 |
| + EFFIINSTRUCT | 0.22 | 1.01 | 36.51 | 1.01 | 15.26 | 1.09 |
| Python 3.11.4 - Intel(R) Xeon(R) Silver 4216 CPU @ 2.10GHz | | | | | | |
| Qwen2.5-Coder-7B | 0.67 | 1.16 | 61.43 | 1.00 | 40.01 | 1.22 |
| +Effi-Code | 0.58 | 1.02 | 60.77 | 0.97 | 32.50 | 1.03 |
| Python 3.11.0 - Intel(R) Xeon(R) Silver 4216 CPU @ 2.10GHz | | | | | | |
| Qwen2.5-Coder-7B | 0.28 | 1.64 | 34.55 | 1.00 | 19.39 | 1.87 |
| + EFFIINSTRUCT | 0.25 | 1.39 | 34.90 | 1.02 | 20.03 | 1.59 |
| Python 3.9.0 - Intel(R) Xeon(R) Silver 4216 CPU @ 2.10GHz | | | | | | |
| Qwen2.5-Coder-7B | 0.30 | 1.60 | 34.26 | 1.01 | 21.02 | 2.10 |
| +Effi-Code | 0.24 | 1.20 | 34.52 | 1.02 | 19.84 | 1.32 |
| Python 3.10.0 - Intel(R) Xeon(R) Silver 4216 CPU @ 2.10GHz | | | | | | |
| Qwen2.5-Coder-7B | 0.29 | 1.63 | 33.26 | 1.01 | 20.32 | 2.16 |
| + EFFIINSTRUCT | 0.26 | 1.43 | 33.50 | 1.02 | 19.53 | 1.61 |

*Table 8.* Evaluation results of EFFICODER effectiveness on different software-hardware setups.

*Table 9.* Performance comparison of different model variants on efficiency metrics.

| Model | ET | NET | MU | NMU | TMU | NTMU |
|---|---|---|---|---|---|---|
| CodeLlama-7b-hf | 1.02 | 0.85 | 42.33 | 1.00 | 23.97 | 0.82 |
| + All | 0.82 (19.6%) | 0.72 (15.3%) | 8.87 (79.0%) | 0.18 (82.0%) | 6.64 (72.3%) | 0.24 (70.7%) |
| + HQ | 1.14 (-11.8%) | 0.98 (-15.3%) | 10.55 (75.1%) | 0.23 (77.0%) | 7.06 (70.5%) | 0.26 (68.3%) |
| + All w/Perf-Cond | 0.92 (9.8%) | 0.81 (4.7%) | 8.91 (79.0%) | 0.19 (81.0%) | 6.99 (70.8%) | 0.25 (69.5%) |
| + HQ + Self-Play | 0.92 (9.8%) | 0.80 (5.9%) | 12.46 (70.6%) | 0.27 (73.0%) | 7.80 (67.5%) | 0.28 (65.9%) |
| + EFFIINSTRUCT | 0.79 (22.5%) | 0.70 (17.6%) | 11.06 (73.9%) | 0.24 (76.0%) | 5.77 (75.9%) | 0.21 (74.4%) |

*Table 10.* Comparison between EffiLearner and EFFIINSTRUCT on the EffiBench dataset, showing efficiency metrics with percentage improvements relative to the baseline model.

| Model | ET | NET | MU | NMU | TMU | NTMU |
|---|---|---|---|---|---|---|
| deepseek-coder-6.7b-instruct | 0.56 | 1.20 | 40.17 | 4.09 | 96.78 | 13.79 |
| + EffiLearner | 0.46 (17.9%) | 0.98 (18.3%) | 40.14 (65.2%) | 1.00 (75.6%) | 15.50 (84.0%) | 1.04 (92.5%) |
| + EFFIINSTRUCT | 0.39 (30.4%) | 0.83 (30.8%) | 40.16 (65.1%) | 1.00 (75.6%) | 15.03 (84.5%) | 0.90 (93.5%) |

all PIE variants on ET and TMU metrics. It achieves a 22.5% reduction in execution time compared to the base model, which is better than even PIE's best "All" strategy (19.6%) in EffiBench. For total memory usage, EFFIINSTRUCT achieves a 75.9% reduction, outperforming all PIE variants.

## A.6. Comparison with EffiLearner

We provide the comparison for EffiLearner and EFFIINSTRUCT fine-tuned DeepSeek-Coder-6.7B-Instruct in Table 10, where we can observe that both EffiLearner and EFFIINSTRUCT improve the efficiency of LLM-generated code. For tasks addressed by all models, average ET decreased from 0.56 (s) to 0.46 (s) with EffiLearner and to 0.39 (s) with EFFIINSTRUCT. For DeepSeek-Coder-6.7B-Instruct, EffiLearner reduced average execution time by 17.9%, while EFFIINSTRUCT achieved a 30.4% reduction.

## A.7. Evaluation results on ENAMEL

The efficiency results between baseline LLMs and EFFIINSTRUCT fine-tuned models on ENAMEL are shown in Table 11, where our results show meaningful improvements in both efficiency and correctness metrics, with effi@1 increasing from 0.373 to 0.458 and pass@1 improving from 0.589 to 0.739 for Qwen2.5-Coder-Instruct-7B.

## A.8. Further discussion with baselines

Compared to existing works (Du et al., 2024; Shypula et al., 2024), EFFIINSTRUCT introduces a fully automated code optimization framework that transforms initial task descriptions into efficient solutions without human intervention. Unlike PIE, which relies on human programmers to write efficient solutions, or Mercury, which selects the most efficient solution from pre-existing human-written code, EFFIINSTRUCT can optimize code starting from just a task description. This automation enables researchers and developers to enhance their existing code generation datasets with minimal manual effort, making efficiency optimization more accessible and scalable. In addition, EFFIINSTRUCT offers broader language

*Table 11.* Performance comparison between baseline LLMs and EFFIINSTRUCT fine-tuned LLMs on the ENAMEL benchmark.

| Model | eff@1 | pass@1 | eff@10 | pass@10 | eff@100 | pass@100 |
|---|---|---|---|---|---|---|
| Qwen2.5-Coder-Instruct-7B | 0.373 | 0.589 | 0.628 | 0.866 | 0.732 | 0.951 |
| + EFFIINSTRUCT | 0.458 | 0.739 | 0.653 | 0.905 | 0.763 | 0.972 |
| DeepSeek-Coder-6.7B-Instruct | 0.179 | 0.299 | 0.549 | 0.822 | 0.727 | 0.922 |
| + EFFIINSTRUCT | 0.393 | 0.654 | 0.633 | 0.887 | 0.752 | 0.937 |

coverage and greater generalizability by including optimized tasks across multiple programming languages (C++, Python, Java, Rust, and Go). This multilingual approach contrasts with PIE's focus on C++ and Mercury's focus on Python, allowing models fine-tuned on EFFIINSTRUCT to perform effectively across diverse language environments. Additionally, EFFIINSTRUCT's significantly larger scale, comprising 65,710 unique tasks compared to Mercury's 1,889 and PIE's 1,474, provides more comprehensive training data, resulting in models demonstrating superior pass@1 rates and efficiency metrics as evidenced in Table 7.

### A.9. Case Study

To illustrate how the source code generated by EFFIINSTRUCT fine-tuned LLM is more efficient than the source code generated by the LLM without fine-tuning on EFFIINSTRUCT, we provide an example in Figure 5. We can observe that the code generated by Qwen2.5-Coder-7B requires 9.89 (s) to execute all unit tests, while the code generated by EFFIINSTRUCT fine-tuned Qwen2.5-Coder-7B with SFT only requires 0.14 (s) to execute. The key reason is that the code generated by Qwen2.5-Coder-7B requires significantly more recursive calls, as it lacks optimized pruning strategies such as breaking early in redundant paths. This inefficiency leads to a much larger number of computations, ultimately resulting in the observed longer execution time. The code generated by EFFIINSTRUCT fine-tuned Qwen2.5-Coder-7B, on the other hand, incorporates smart optimizations, such as terminating recursion early when certain conditions are met, thereby reducing the overall time complexity.

---

**Task Description**

**Problem:** You are given an integer array `cookies`, where `cookies[i]` denotes the number of cookies in the `ith` bag. You are also given an integer `k` that denotes the number of children to distribute `all` the bags of cookies to. All the cookies in the same bag must go to the same child and cannot be split up. The `unfairness` of a distribution is defined as the `maximum total` cookies obtained by a single child in the distribution. Return the `minimum` unfairness of all distributions.
solution = Solution() assert solution.distributeCookies([8, 15, 10, 20, 8], 2) == 31 assert solution.distributeCookies([6, 1, 3, 2, 2, 4, 1, 2], 3) == 7

```
solution = Solution()
assert solution.distributeCookies([8, 15, 10, 20, 8], 2) == 31
assert solution.distributeCookies([6, 1, 3, 2, 2, 4, 1, 2], 3) == 7
```

**Qwen2.5-Coder-7B**

```
Execution Time: 9.89 (s)
Max Memory Peak: 24.39 (MB)
Memory Usage: 240.47 (MB*s)
class Solution:
    def distributeCookies(self, cookies, k):
        min_unfairness = float('inf')
        distribution = [0] * k
        def distribute(i):
            nonlocal min_unfairness
            if i == len(cookies):
                min_unfairness = min(min_unfairness,\
                max(distribution))
                return
            for j in range(k):
                distribution[j] += cookies[i]
                distribute(i + 1)
                distribution[j] -= cookies[i]
        distribute(0)
        return min_unfairness
```

**Qwen2.5-Coder-7B SFT with EFFIINSTRUCT**

```
Execution Time: 0.14 (s)
Max Memory Peak: 24.39 (MB)
Memory Usage: 2.47 (MB*s)
class Solution:
    def distributeCookies(self, cookies, k):
        def backtrack(i):
            nonlocal ans
            if i == len(cookies):
                ans = min(ans, max(children))
                return
            for j in range(k):
                children[j] += cookies[i]
                backtrack(i + 1)
                children[j] -= cookies[i]
                if children[j] == 0:
                    break
        children = [0] * k
        ans = float('inf')
        backtrack(0)
        return ans
```

*Figure 5.* A case illustration for the task with code generated by Qwen2.5-Coder-7B and EFFIINSTRUCT fine-tuned Qwen2.5-Coder-7B in EffiBench `problem_idx=2305`.

*Table 12.* Code efficiency and pass@1 of Qwen2.5-Coder-7B with EFFIINSTRUCT with the five times execution on EffiBench.

| Model | ET (s) ↓ | NET ↓ | MU (Mb) ↓ | NMU ↓ | TMU (Mb*s) ↓ | NTMU ↓ |
|---|---|---|---|---|---|---|
| Random Execution 1 | 0.17 | 1.30 | 32.71 | 1.03 | 8.31 | 2.23 |
| Random Execution 2 | 0.17 | 1.31 | 32.93 | 1.04 | 8.35 | 2.28 |
| Random Execution 3 | 0.17 | 1.30 | 32.71 | 1.03 | 8.23 | 2.22 |
| Random Execution 4 | 0.17 | 1.30 | 32.84 | 1.04 | 8.30 | 2.25 |
| Random Execution 5 | 0.17 | 1.30 | 32.88 | 1.04 | 8.28 | 2.27 |
| mean | 0.17 | 1.302 | 32.814 | 1.037 | 8.293 | 2.249 |
| std | 0.0 | 0.003 | 0.09 | 0.003 | 0.038 | 0.023 |

## A.10. Randomness

To ensure reliable model performance, we also account for variability in system conditions. Metrics like Execution Time (ET), Max Memory Usage (MU), and Total Memory Usage (TMU) might fluctuate due to factors like server workload and hardware availability, introducing noise that affects performance measurements. To demonstrate whether our results are affected by such randomness, we provide five results at different times with the mean and std for Qwen2.5-Coder-7B fine-tuned with EFFIINSTRUCT in Table 12. We can observe that the results are robust as the std of the five execution times is very low for all metrics. For example, the std of ET for the five executions is 0.00.

