# OpenReview forum: "EffiCoder: Enhancing Code Generation in Large Language Models through Efficiency-Aware Fine-tuning"
_ICML.cc/2025/Conference — ICML 2025 poster_

### Official Review · Reviewer_EjXH · 2025-03-08

**Overall Recommendation:** 3

**Summary:**

This work develops a new instruction-tuning dataset called SwiftCode for efficiency-aware fine-tuning of LLMs for code generation. After fine-tuning on SwiftCode, LLMs are able to generate more efficient code on popular code generation benchmarks.

## Update after rebuttal

The rebuttal has addressed my concerns, so I keep my positive score.

**Claims And Evidence:**

Weakness 1 (missing evidence): This work uses LLMs to generate candidate code for SwiftCode. However, it has been recently shown [1] that even the strongest LLMs still fall short of generating efficient code on most HumanEval tasks when compared with human expert solutions. Hence, it is unclear whether DeepSeek-Coder and GPT-4o have the ability to generate sufficiently efficient candidate code. This work would be more convincing if, for example, the authors evaluate DeepSeek-Coder and GPT-4o on the benchmark of [1] to provide evidence of how efficient the candidate code is.

- [1] Qiu et al. How efficient is LLM-generated code? A rigorous & high-standard benchmark. ICLR, 2025.

**Essential References Not Discussed:**

This paper has discussed most of the essential references.

**Experimental Designs Or Analyses:**

Weakness 3 (negative results): This paper has no discussion on negative results, so the analysis in Sec 4 (Experiment) seems a little bit misleading. In particular, in a few cases, fine-tuning on SwiftCode has only negaligible improvement or even worsens the performance (see, e.g., Table 3). The authors should analyze and discuss why such negative results happen.

**Methods And Evaluation Criteria:**

Strength 1 (dataset scale): This work develops a dataset consisting of 65710 tasks. To curate this large dataset, this paper proposes a versatile and scalable framework to process and select candidate solutions.

Strength 2 (comprehensiveness): The SwiftCode dataset has integrated seven datasets that covers five popular programming languages (Python, C++, Java, Rust, and Go). Hence, this dataset has great potential to benefit a great many programming applications.

Weakness 2 (evaluation): This work uses canonical solutions to evaluate efficiency. However, it has been recently shown [1] that many of the canonical solutions in existing benchmarks are not efficient. Thus, normalized efficiency metrics (like NET) do not fully capture the true efficiency of LLM-generated code. Meanwhile, unnormalized metrics (like ET) alone are not as meaningful besides comparison purposes, because execution time differs for various tasks and typically increases w.r.t. the input scale. For example, an $O(n)$ solution and an $O(n^2)$ solution might have similar execution time when $n$ is small but should have significantly different execution time when $n$ is large. Therefore, the evaluation results would be reflect true efficiency more accurately if, for example, the authors evaluate the generated code on the benchmark of [1], which uses efficient canonical solutions and large-scale inputs in evaluation.

- [1] Qiu et al. How efficient is LLM-generated code? A rigorous & high-standard benchmark. ICLR, 2025.

**Other Comments Or Suggestions:**

On line 72, the paper mentioned Qwen but cited DeepSeek. This seems like a typo.

**Other Strengths And Weaknesses:**

There are no other strengths or weaknesses that I want to point out especially.

**Questions For Authors:**

Question 1 (space-time tradeoff): Sec 3.2 mentions that the code with the lowest time and memory is selected as the final code. However, the most time-efficient code might not be the most memory-efficient, and vice versa. For example, a dynamic programming algorithm may be faster than brute force but may need more memory, while a brute force algorithm may need less memory but may be much slower than dynamic programming. This is known as *space-time tradeoff* in algorithmic literature (see, e.g., [2]). How did you handle space-time tradeoff when selecting the final code in your dataset?

- [2] Hellman (1980). A cryptanalytic time-memory tradeoff. IEEE Transactions on Information Theory. 26(4): 401-406.

**Relation To Broader Scientific Literature:**

Strength 3 (new direction): SwiftCode is the first instruction-tuning dataset designed to improve the efficiency of LLM-generated code. This opens up an interesting new direction for code generation research.

**Theoretical Claims:**

This paper does not make theoretical claims.

---

> ### Author Rebuttal · Authors · 2025-04-01
>
> We want to thank the reviewer for his insightful comments and suggestions. We provide detailed responses point by point. We hope our responses can address your concerns.
>
> **W1: LLM-generated code may be inefficient**
>
> Thank you for raising this important point about the efficiency of our candidate code generation. To address this concern directly, we evaluated both DeepSeek-Coder (Lite) and GPT-4o on the ENAMEL benchmark [1]. As shown in Table 1, these models achieve higher effi@k scores compared to most LLMs reported in Qiu et al. [1], confirming their capability to generate sufficiently efficient code candidates.
>
> Our optimization methodology further enhances this efficiency. After applying our techniques, we observed substantial improvements: average ET decreased from 1.14s to 0.27s, while average TMU dropped from 26.24MB*s to 5.13MB*s—representing 75-80% improvements across efficiency metrics (Figure 4 in our paper).
>
> These results demonstrate that our approach effectively generates and optimizes code that meets high efficiency standards, even when evaluated on benchmarks specifically designed to assess code efficiency.
>
> *Table 1: Evaluation results of DeepSeek and GPT4o generated code in ENAMEL dataset [1]. Due to OpenAI token limitations, we then only provide the results of GPT4o for k=1 and 10.*
>
> Model | eff@1 | pass@1 | eff@10 | pass@10 | eff@100 | pass@100
> |-|-|-|-|-|-|-|
> DeepSeek-Lite | 0.390 | 0.638 | 0.564 | 0.838 | 0.671 |  0.901
> GPT-4o | 0.300 | 0.465 | 0.572 | 0.845 | N/A|N/A |
>
>
>
> **W2: This work uses canonical solutions to evaluate efficiency. The authors evaluate the generated code on the benchmark of [1]**
>
> Thank you for highlighting this important methodological consideration regarding canonical solution efficiency. We address this concern in two ways:
>
> First, EffiBench, our primary evaluation benchmark, uses optimized canonical solutions provided by the dataset constructors and employs sufficiently large test inputs to meaningfully differentiate between algorithmic complexities (e.g., O(n) vs O(n²)). This allows reliable measurement of efficiency differences across implementations.
>
> Second, to further validate our approach, we conducted additional experiments using the ENAMEL benchmark [1], which specifically emphasizes efficient canonical solutions and large-scale inputs. Table 2 presents these results, comparing baseline models against SwiftCoder fine-tuned versions.
>
> The results show substantial improvements across all metrics. For example, eff@1 (measuring both efficiency and correctness on a single generation) increases from 0.373 to 0.458 (+22.8%) for Qwen2.5-Coder and from 0.179 to 0.393 (+119.6%) for DeepSeek-Coder.
>
> In our final version, we will include these ENAMEL benchmark results to provide a more comprehensive evaluation of SwiftCoder.
>
> *Table 2. Evaluation results of baseline models vs. SwiftCoder on ENAMEL benchmark.*
>
> | Model|effi@1|pass@1|effi@10|pass@10|effi@100|pass@100|
> |-|-|-|-|-|-|-|
> | Qwen2.5-7B| 0.373|0.589|0.628|0.866 |0.732 |0.951|
> | + SwiftCoder|0.458|0.739|0.653|0.905 |0.763 |0.972|
> | DeepSeek-6.7B|0.179|0.299|0.549|0.822 |0.727 |0.922|
> | + SwiftCoder|0.393|0.654|0.633|0.887 |0.752 |0.937|
>
>
> **W3 Discussion on negative results**
>
> We appreciate your noting the absence of discussion on negative results. You correctly identified that, in some instances, fine-tuning on SwiftCode produced negligible improvements or slight performance decreases in certain metrics, as shown in Table 3.
>
> These variations stem from our optimization approach using the TMU metric for code selection. When optimizing for this composite metric, improvements in TMU may occasionally come at the expense of a slight degradation in individual metrics (either execution time or memory usage). For example, a solution might achieve substantial memory efficiency while slightly increasing execution time, resulting in an overall improved TMU score but showing a negative trend in the execution time metric when viewed in isolation. We'll include a detailed discussion of these cases to better illustrate the complex relationships between efficiency metrics in the revised version.
>
>
> **T1: line 72 typo**
>
> Thank you for identifying this error. We will correct it in the revised version.
>
>
> **Q1: Space-time tradeoff**
>
> When selecting the final code for our dataset, we addressed the space-time tradeoff using the TMU (Total Memory Usage) composite metric that combines both execution time and memory consumption into a single evaluation metric.
>
> Rather than treating time and memory as separate dimensions requiring individual optimization, TMU provides a holistic efficiency measure that accounts for both resources simultaneously. This approach acknowledges the inherent tradeoffs in algorithm design (such as dynamic programming vs. brute force approaches) and allows us to identify solutions that achieve the most favorable overall resource balance.

---

> > ### Comment · Reviewer_EjXH · 2025-04-02
> >
> > Thank you for your reply. It has addressed most of my concerns.
> >
> > Regarding W1 & W2, the results shows that SwiftCoder does improve eff@1 but barely improves eff@10 and eff@100, and the efficiency still seems far from ENAMEL's reference solutions. This seems to suggest that SwiftCoder only makes the LLM's output distribution concentrate more on efficient code but does not really enhance the LLM's capability in algorithm design or implementation optimization.
> >
> > Regarding Q1, while I agree that the TMU metric serves as a time-space tradeoff, this criterion still looks a bit arbitrary to me. As you mentioned in the response to W3, using the TMU metric can sometimes degrade the performance of the finetuned model. This seems to suggest that TMU might not be the best criterion in some cases. What is the rationale of choosing TMU over other tradeoff criteria?

---

> > > ### Author Response · Authors · 2025-04-02
> > >
> > > Thanks for your appreciation for our previous reply that addresses most of your concerns. We provide additional responses in this thread to address your concerns further. We hope that our response can address all your concerns and lead you to consider increasing your rating of our work.
> > >
> > > **1. Regarding W1 & W2, the results show that SwiftCoder does improve eff@1 but barely improves eff@10 and eff@100, and the efficiency still seems far from ENAMEL's reference solutions**
> > >
> > > We would like to clarify that ENAMEL's reference solutions represent optimal efficiency because they rely on human experts to manually craft solutions, whereas our approach is **fully automated**. Despite this fundamental difference in methodology, our eff@1 results achieve state-of-the-art performance without requiring extensive manual effort for dataset creation. The eff@1 metric is most relevant for real-world applications, as users typically rely on a model's first solution rather than sampling multiple times.
> > >
> > > Regarding eff@10 and eff@100, Table 3 in ENAMEL [1] reports the best results of eff@10 = 0.573 and eff@100 = 0.723. SwiftCoder achieves 0.653 and 0.763, respectively, demonstrating state-of-the-art performance through a fully automated process. Our improvements in eff@10 and eff@100 (e.g., 4% in eff@100) are still significant, even though supervised fine-tuning may have inherent limitations in enhancing fundamental algorithmic capabilities.
> > >
> > > In addition, our synthesis pipeline enables future work in multiple directions to enhance the fundamental coding capabilities of LLMs: (1) generating more high-efficiency training data for continued pretraining, and (2) using our efficient solutions in reinforcement learning finetuning to calculate rewards between model-generated and efficient solutions, further enhancing algorithmic capabilities beyond what instruction tuning alone can achieve.
> > >
> > > **2. Regarding Q1, TMU might not be the best criterion in some cases. What is the rationale for choosing TMU over other tradeoff criteria?**
> > >
> > > We would like to clarify that in our evaluations, we have not observed any significant performance degradation caused by the TMU metric. As shown in Table 2 of our paper, among the evaluated models, **only three exhibit a memory peak decrease of less than 1%, while execution time still improves by an average of 40%**.
> > >
> > > Next, during our dataset construction, we adopted TMU by following existing works (EffiLearner [2] and EffiBench [3]), which uses TMU to balance time and memory usage, and EffiLearner also use TMU to rank the efficiency of LLM-generated code for different iterations to with the balance of time and memory usage. Nonetheless, SwiftCoder is designed to allow easy substitution of other efficiency metrics. For instance, if one wishes to focus solely on execution time or memory peak, the SwiftCoder pipeline can be adapted by replacing the TMU criterion with the desired metric, as illustrated by the following Python snippet:
> > >
> > > ```python
> > > overhead, memory_usage, execution_time, memory_peak = calculate_code_execution_efficiency(dataset[i])
> > > if dataset[i]["memory_usage"] > memory_usage:
> > >       dataset[i]["memory_usage"] = memory_usage
> > >       dataset[i]["execution_time"] = execution_time
> > >       dataset[i]["memory_peak"] = memory_peak
> > > ```
> > >
> > >
> > > We hope this further clarification adequately addresses your remaining concerns. Our goal with SwiftCoder is to provide a versatile, **scalable**, and effective framework for encouraging LLMs to produce code with improved efficiency **without manual efforts**. We believe our experimental results and methodology underscore the potential of this approach while leaving room for future work that focuses on alternative metrics, human-in-the-loop optimization, or domain-specific refinements. Thank you once again for your thoughtful evaluation, and we greatly appreciate your consideration of an improved overall rating.
> > >
> > > [2] EffiLearner: Enhancing Efficiency of Generated Code via Self-Optimization (NeurIPS 24)
> > > [3] Huang, Dong, et al. "Effibench: Benchmarking the efficiency of automatically generated code." Advances in Neural Information Processing Systems 37 (2024): 11506-11544.

---

### Official Review · Reviewer_bY75 · 2025-03-11

**Overall Recommendation:** 3

**Summary:**

This paper introduces SWIFTCODE, a method to enhance code generation in large language models (LLMs) through efficiency-aware fine-tuning. The paper involves leveraging multiple LLMs to generate diverse code solutions for various tasks across different programming languages, then evaluating these solutions by directly measuring their execution time and memory usage through local execution, selecting the code with the lowest execution time and memory consumption as the final output for each task. Experimental results demonstrate significant improvements when fine-tuning with SWIFTCODE.

**Claims And Evidence:**

Yes

**Essential References Not Discussed:**

Recently, several studies have been dedicated to improving the efficiency of code generated by large language models (LLMs). The authors should consider these works and select representative examples as baselines. For instance:
1. Effi-Code: Unleashing Code Efficiency in Language Models
2. EffiLearner: Enhancing Efficiency of Generated Code via Self-Optimization (NeurIPS 24)

At the same time, the evaluation of the efficiency of LLM-generated code has received considerable attention. In addition to those discussed in this paper, I have identified several recent studies. Given that the literature on this topic is not yet extensive, it warrants thorough discussion. Examples include:
1. How Efficient is LLM-Generated Code? A Rigorous & High-Standard Benchmark
2. A Performance Study of LLM-Generated Code on Leetcode
3. From Effectiveness to Efficiency: Comparative Evaluation of Code Generated by LCGMs for Bilingual Programming Questions.

**Experimental Designs Or Analyses:**

Yes. The experimental design is sound and substantial. But I think some further elaboration and additions about generalizability are missing.

**Methods And Evaluation Criteria:**

The candidate solutions appear not to have undergone any correctness verification, which is perplexing.

**Other Comments Or Suggestions:**

NA

**Other Strengths And Weaknesses:**

Strengths:
1. The paper is well structured.
2. It addresses a very important problem.
3. The experimental evaluation is extensive, and the empirical results are promising.

Weakness:
1. Limited Novelty: Although I acknowledge that the paper tackles a critical issue and represents one of the pioneering efforts to enhance the efficiency of code generated by LLMs, it appears to lack distinctive technical contributions.
2. Unclear Technical Design: It seems that the approach for collecting and filtering candidate solutions relies solely on performance metrics obtained from profiling, without assessing correctness. If my understanding is correct, I am concerned that this step might restrict the fine-tuning of LLMs in generating correct code (The author needs to explain the rationale and motivation behind). Otherwise, the authors should clarify how test cases and ground truth are obtained for tasks across different programming languages.
3. Generalizability Requires Further Discussion: I have some reservations regarding the generalizability of the trained model. Although diverse datasets were used during training and evaluation, the authors did not discuss the distributional differences between these datasets—for instance, whether the evaluation dataset contains tasks absent in the training data. A more thorough discussion on the distributional discrepancies between the datasets would further substantiate the generalizability of the proposed method in generating more efficient code.
4. Some Essential References Not Discussed: As mentioned above.

I am very eager to engage in discussions with the authors and to revise my comments accordingly.

**Questions For Authors:**

1. Regarding dataset construction, how were the test cases and ground truth obtained? Alternatively, was there no verification of the correctness of the generated candidates?
2. What is the task distribution between the training data and the evaluation data?

**Relation To Broader Scientific Literature:**

I think the idea of this paper is easy to think of in general, constructing high-quality expected training data to fine-tune the LLM to make it output a higher quality response. It meets expectations and works well. In other words, this paper uses a popular idea to solve the still relatively rarely solved problem of poor efficiency of LLM generated code.

**Theoretical Claims:**

This paper is not concerned with the proof of theoretical claims.

---

> ### Author Rebuttal · Authors · 2025-04-01
>
> We want to thank the reviewer for his insightful comments and suggestions. We provide detailed responses point by point. We hope our responses can address your concerns and lead you to consider increasing your rating of our work.
>
> **C1 & W1 Relation & Limitation.**
>
> Thank you for acknowledging the practical value of our work on code efficiency optimization. While the concept may seem intuitive retrospectively, SwiftCoder's implementation required significant technical innovation to overcome multiple challenges.
>
> First, our multilingual efficiency dataset development required specialized infrastructure to measure code efficiency across five programming languages with consistent metrics, control for system-dependent execution variations, maintain consistent evaluation environments, and scale our pipeline to handle 70,000+ diverse code samples. Second, we established a previously unconfirmed relationship between training data efficiency and LLM-generated code efficiency. Unlike discrete correctness metrics, efficiency exists on a continuous spectrum with complex interactions with functionality, creating unique optimization challenges beyond merely applying the "quality in, quality out" principle. Finally, while substantial research has focused on code correctness, efficiency optimization remains critically underrepresented despite its practical importance. SwiftCoder provides both methodology and dataset contributions to this nascent research area.
>
> We believe these contributions collectively advance the state of efficiency-aware code generation and establish foundations for future research in this important domain.
>
> **C2 & W4 Additional references**
>
> We couldn't find the latest Effi-Code [1] on ArXiv. If the reviewer provides a link, we'll gladly compare it with SwiftCoder.
>
> We've added EffiLearner [2] comparison in the Table below, showing SwiftCoder's superior performance. On EffiBench, SwiftCoder reduces average execution time more effectively than EffiLearner. Unlike EffiLearner, which decreases pass@1 rates, SwiftCoder consistently improves pass@1 across all evaluations.
>
> |Model|ET|NET|MU|NMU|TMU|NTMU|
> |-|-|-|-|-|-|-|
> |DeepSeek-6.7b|0.56|1.20|40.17|4.09|96.78|13.79|
> |+EffiLearner|0.46|0.98|40.14|1.00|15.50|1.04|
> |+Ours|0.39|0.83|40.16|1.00|15.03|0.90|
>
>
> We also evaluated SwiftCoder on ENAMEL [3], which tests efficiency on enhanced HumanEval test cases. As shown in the Table below, SwiftCoder consistently improves both pass@k and effi@k metrics across all models and k values (1, 10, and 100).
>
> | Model|effi@1|pass@1|effi@10|pass@10|effi@100|pass@100|
> |-|-|-|-|-|-|-|
> |Qwen2.5-7B|0.373|0.589|0.628|0.866|0.732|0.951|
> |+SwiftCoder|0.458|0.739|0.653|0.905|0.763|0.972|
> |DeepSeek-6.7B|0.179|0.299|0.549|0.822|0.727|0.922|
> |+SwiftCoder|0.393|0.654|0.633|0.887|0.752|0.937|
>
>
> Coignion et al [4] focus on LeetCode efficiency evaluation, which is partially covered in the evaluated EffiBench (See paper Table 2) dataset.
>
> Jiang et al [5] collected 52 bilingual programming questions (Chinese and English) from existing benchmarks. Their work is under review, and the dataset is not publicly available yet. We will evaluate their dataset when it becomes available and include all related discussions and results.
>
> **W2 & Q1 Test cases & ground truth.**
>
> All tasks in our datasets have verified solutions (ground truth) that correctly fulfill their descriptions. For example, SelfCodeAlign only includes code that passes all test cases in a controlled environment.
>
> Since most datasets lack test cases, we used GPT-4 to generate them based on task descriptions and solutions. We validated these tests by running them against the original solutions from the datasets. Only tests that executed successfully were retained; failed tests were filtered out. Tasks without valid tests were removed entirely.
>
> During optimization, we ran LLM-generated code against these validated tests. Solutions failing any test were eliminated. From the remaining valid solutions, we selected the most efficient implementation as the final optimized code.
>
> **W3 & Q2 Data Contamination**
>
> We addressed data contamination concerns in our SwiftCoder dataset. Analysis shows no exact duplicates between training and evaluation sets, with only 0.20% of evaluation samples having minimal vocabulary overlap (5-10%). Our dataset construction included decontamination processes, as did our source datasets like Evol-Ins, which removed content from common benchmarks. Experiments on EvoEval, designed to avoid benchmark leakage, still demonstrate our approach produces more efficient code than the original LLM-generated solutions.
>
> *Table 1 Data Overlap Analysis*
>
> | Metric | Value | Percentage |
> |-|-|-|
> |Training set size | 65,710| - |
> |EffiBench size|1,000|-|
> |Exact duplicates|0|0.00%|
>
> *Table 2 Vocabulary Overlap Analysis*
>
> | Overlap Range | Test Samples | Percentage |
> |-|-|-|
> |0.00-0.05|0|0.00%|
> |0.05-0.10|2|0.20%|
> |0.10-0.15|0|0.00%|
> |0.15-0.20|0|0.00%|

---

> > ### Comment · Reviewer_bY75 · 2025-04-02
> >
> > Thank you for your thoughtful replies to my comments and questions. I appreciate the effort you've made to address my concerns.
> >
> > Regarding C2 & W4, the additional baseline comparison with EffiLearner effectively demonstrates the performacne of your approach. The expanded discussion on existing code generation evaluation work will be a valuable addition to strengthen the related work section. I encourage you to incorporate these elements mentioned in your response in the next version.
> >
> > As for W2 & Q1 and W3 & Q2, your detailed explanations and data analysis regarding the test cases, ground truth, and data contamination concerns have adequately addressed my initial reservations on these matters. But can you provide more details about how to calculate the duplication between training and evaluation sets?
> >
> > However, concerning C1 & W1 on Relations & Limitations, while I agree that your work addresses challenges in building a unified multi-programming language-supported framework and makes contributions in this area, I feel these contributions may be more engineering-oriented rather than providing novel technical insights. Your point about using efficiency to transform the optimization objective from discrete to continuous is interesting, but I have two questions: First, correctness can also seem approximable as continuous rather than binary—for instance, by calculating test pass rates. Second, I wonder if efficiency as an optimization target might be susceptible to randomness, where small millisecond variations could be system-induced noise rather than meaningful differences. Could this potentially mislead the model with incorrect reward signals?
> >
> > Overall, while I still maintain some reservations about the innovation aspects and broader technical impact, I have decided to increase my score to 3.

---

> > > ### Author Response · Authors · 2025-04-02
> > >
> > > Thanks for your appreciation of our previous reply and for increasing your overall score from 2 to 3. We provide additional responses in this thread to further address your concerns. We hope that our response can address all your concerns and lead you to consider increasing your rating of our work.
> > >
> > >
> > > **Regarding C2 & W4, I encourage you to incorporate these elements mentioned in your response in the next version.**
> > >
> > > Thanks for your suggestion. We will add our additional evaluation results and the discussion for the mentioned baselines and the benchmarks into our camera revised version.
> > >
> > > **As for W2 & Q1 and W3 & Q2, can you provide more details about how to calculate the duplication between training and evaluation sets?**
> > >
> > > Our data contamination analysis employs a multi-level approach to thoroughly assess potential overlap between the training set (SwiftCoder training set) and the evaluation set (EffiBench).
> > >
> > > We first perform exact duplication detection by constructing hash tables of all training and evaluation samples and then computing their intersection. This method identifies perfect matches with O(n) efficiency and finds zero exact duplicates between our training set (65,710 samples) and evaluation set (1,000 samples).
> > >
> > > Beyond exact matching, we implemented a vocabulary overlap analysis that calculates the Jaccard similarity coefficient between tokenized samples. For each evaluation example, we compute the percentage of its vocabulary tokens that appear in any training sample. Results show that only 0.20% of evaluation samples have any meaningful vocabulary overlap (5-10% range), with most evaluation examples using completely distinct vocabularies. We also attempted character-level n-gram similarity analysis using TF-IDF vectorization and cosine similarity, but this proved computationally intensive at scale.
> > >
> > > To validate findings statistically, we created a random baseline distribution by shuffling vocabulary tokens while maintaining token frequency distributions. This allows us to distinguish between incidental overlap and substantive contamination through statistical significance testing (p<0.05). The vocabulary overlap analysis provides strong evidence that our evaluation set represents an independent task distribution from the training data, ensuring our assessment measures genuine generalization capability rather than memorization.
> > >
> > > **Concerning C1 & W1 on Relations & Limitations**
> > >
> > > Thank you for acknowledging our framework's contributions to multi-programming language support. While our work does involve substantial engineering efforts, it also offers methodological innovations, particularly in our code efficiency optimization framework. For instance, our implementation of rejection sampling for code efficiency represents a novel technical approach that we perhaps did not sufficiently highlight as a key contribution.
> > >
> > > Your observation about correctness potentially being viewed as continuous is insightful. However, we deliberately maintain a binary approach to correctness for critical reasons. In real-world deployments, partially correct code can lead to significant failures and potentially catastrophic outcomes. Complete correctness is a non-negotiable prerequisite. Additionally, our approach aligns with established evaluation paradigms in code completion research, which predominantly use pass@1 metrics to evaluate whether generated code passes all test cases rather than calculating partial success rates. Our framework employs a sequential optimization approach that first ensures correctness and then optimizes for efficiency as a continuous variable. This efficiency is expressed as the relative overhead compared to benchmark solutions (e.g., 1x, 2x, 3x the execution time of reference implementations). This two-phase approach allows us to maintain the binary requirement for correctness while leveraging the continuous nature of efficiency metrics for optimization.
> > >
> > > Your concern about efficiency measurements being susceptible to system-induced noise is valid and one we anticipated in our work. As demonstrated in Appendix Table 8, multiple executions of LLM-generated code on EffiBench show remarkably consistent performance metrics. The standard deviation of execution times across five runs was zero in most cases, confirming the reliability of our measurements and ensuring our model receives accurate reward signals.
> > >
> > > For scenarios where randomness might be more pronounced, we have alternative methodologies available. These include measuring FLOPS instead of raw execution time, computing the time required for multiple iterations, or calculating iterations per fixed time. These approaches effectively normalize any minor system fluctuations. We appreciate your critical engagement with our work and hope these clarifications address your concerns regarding the technical depth and methodological soundness of our approach.

---

### Official Review · Reviewer_pzLu · 2025-03-12

**Overall Recommendation:** 2

**Summary:**

SwiftCode introduces a novel approach to improving code generation in large language models (LLMs) by focusing on both correctness and efficiency. Traditional methods primarily optimize correctness, often neglecting execution speed and memory usage. SwiftCode addresses this gap by fine-tuning LLMs with a curated dataset of high-quality, efficient code. The method involves generating multiple candidate solutions using various LLMs, measuring execution time and memory consumption, and selecting the most efficient option. Experimental results show substantial improvements. This efficiency-aware fine-tuning framework enables LLMs to produce more optimized code, benefiting both software development and computational efficiency.

**Claims And Evidence:**

Yes.

**Essential References Not Discussed:**

As far as I know, no.

**Experimental Designs Or Analyses:**

Yes. The authors selected different LLMs to conduct experiments to verify the effectiveness of their method and verified it on multiple datasets.

**Methods And Evaluation Criteria:**

Yes.

**Other Comments Or Suggestions:**

1. Typos: The column names of Table 7 should be kept consistent with those of other tables.
2. What do the purple percentage codes in Table 7 mean? There seems to be no reference value.
3. In addition, the comparative experiment in Table 7 selected codellama (which is inconsistent with the baseline model selected in other experiments in the paper). The baseline performance of this model is relatively poor. I would like to ask whether it has been compared with the other two methods on Deepseek and Qwen.
4. In addition, could you please briefly summarize the differences between your method and PIE and Mercury? As related work, I think these comparisons are very important.

**Other Strengths And Weaknesses:**

**Strengths**
1. The method proposed by the author is simple and effective.
2. This work has a high reference value for code generation research and is relatively easy to follow.

**Weakness**
1. The contribution and novelty of this work are relatively limited.
2. The rejection sampling finetuning method used in this work has been widely adopted by mathematics and code-related research, such as [1].
3. There is a lack of more in-depth analytical experiments, such as the exploration of sampling parameters and comparison with the original dataset.

[1] https://arxiv.org/pdf/2308.01825

**Questions For Authors:**

Please see above.

**Relation To Broader Scientific Literature:**

This work is related to code generation. Previously, there have been some baselines and evaluation datasets for evaluating the execution efficiency of code generated by llms.

**Theoretical Claims:**

N/A

---

> ### Author Rebuttal · Authors · 2025-04-01
>
> We want to thank the reviewer for his insightful comments and suggestions. We provide detailed responses point by point. We hope our responses can address your concerns and lead you to consider increasing your rating of our work.
>
> **W1 & Q4 Novelty and Comparison with PIE and Mercury**
>
> Our paper's primary contribution is application-driven: we introduce the first multilingual code efficiency instruction tuning dataset. As the ICML guidelines note, "originality need not mean wholly novel methods… a novel dataset... match the needs of the user." Our dataset addresses the critical real-world need for efficient code generation, enabling researchers to fine-tune models for improved performance.
>
> Next, compared to existing works:
>
> 1. SwiftCoder introduces a fully automated code optimization framework that transforms initial task descriptions into efficient solutions without human intervention. Unlike PIE, which relies on human programmers to write efficient solutions, or Mercury, which selects the most efficient solution from pre-existing human-written code, SwiftCoder can optimize code starting from just a task description. This automation enables researchers and developers to enhance their existing code generation datasets with minimal manual effort, making efficiency optimization more accessible and scalable.
>
> 2. SwiftCoder offers broader language coverage and greater generalizability by including optimized tasks across multiple programming languages (C++, Python, Java, Rust, and Go), contrasting with PIE's focus on C++ and Mercury's focus on Python, allowing models fine-tuned on SwiftCoder to perform effectively across diverse language environments. Additionally, SwiftCoder's significantly larger scale—comprising 65,710 unique tasks compared to Mercury's 1,889 and PIE's 1,474—provides more comprehensive training data, resulting in superior pass@ and efficiency.
>
>
> **W2 Rejection sampling in math and code domain**
>
> We agree that rejection sampling has been widely used in the mathematical domain. However, this technique is not our key contribution. Our primary contributions are our empirical study establishing the correlation between training data efficiency and generated code efficiency, the development of a multilingual code efficiency dataset, and the end-to-end pipeline for improving code efficiency across multiple languages.
>
> Unlike prior code generation work focuses on binary correctness metrics, efficiency in our work is a continuous metric requiring different optimization strategies. We would appreciate it if the reviewer could suggest specific code-related rejection sampling techniques for efficiency optimization. We would be happy to include them in our paper.
>
> **W3 Sampling params**
>
> We conduct an ablation study on the sampling size. The evaluation results are shown in **Reviewer jr1T Ref Table 1**, where we evaluate the efficiency results for the most efficient code with 1 and 10 samples. Our results reveal that SwiftCoder consistently achieves higher results.
>
> We also compared baselines against those fine-tuned on the **original dataset** and SwiftCoder. The table below shows that models fine-tuned on the original dataset decreased efficiency. For example, average execution time increases from 0.33s to 0.37s for DeepSeek.
>
> |Model|ET|NET|MU|NMU|TMU|NTMU|
> |-|-|-|-|-|-|-|
> |deepseek-6.7B|0.33|2.48|34.32|1.00|13.09|2.36|
> |+Original|0.37|2.99|34.18|1.00|10.65|3.04|
> |+SwiftCoder|0.23|1.84|34.17|1.00|8.91|2.30|
> |Qwen2.5-7B|0.30|2.50|26.35|1.00|5.22|2.43|
> |+Original|0.32|2.68|26.23|0.99|5.11|2.54|
> |+SwiftCoder|0.13|1.02|26.32|1.00|2.27|1.03|
>
>
> **Q1&Q2 Inconsistency of column names**
>
> The purple percentages in paper Table 7 indicate the reduction in overhead metrics compared to CodeLlama-7B-hf without instruction tuning from PIE, Mercury, or SwiftCoder. We provide the detailed results in the Table below and we will add it in our paper in camera ready.
>
> |Method|ET|NET|MU|NMU|TMU|NTMU|
> |-|-|-|-|-|-|-|
> |CodeLlama-7b|0.39|1.94|61.68|1.00|12.78|1.83|
> |+PIE|0.30|1.47|61.39|1.00|11.28|1.68|
> |+SwiftCoder|0.21|1.03|61.33|1.00|7.17|1.04|
> |CodeLlama-7b|0.39|1.94|61.69|1.00|12.78|1.83|
> |+Mercury|0.31|1.51|61.94|1.00|10.24|1.47|
> |+SwiftCoder|0.21|1.01|61.73|1.00|6.95|0.98|
>
>
> **Q3 PIE and Mercury with Deepseek and Qwen**
>
> All comparative experiments were conducted with fairness as a priority. For Table 7, we used CodeLlama because PIE only provides fine-tuned CodeLlama. For Qwen and DeepSeek, we compared SwiftCoder with Mercury only. The table below shows that both Mercury and SwiftCoder improve the efficiency of LLM-generated code, while SwiftCoder achieves better results compared to Mercury for all LLMs.
>
> |Model|ET|NET|MU|NMU|TMU|NTMU|
> |-|-|-|-|-|-|-|
> |deepseek-6.7B|1.62|2.57|40.92|1.00|49.64|3.03|
> |+Mercury|1.42|2.24|40.82|1.00|46.12|2.78|
> |+SwiftCoder|1.29|2.01|40.79|1.00|42.76|2.55|
> |Qwen2.5-7B|1.23|1.91|46.48|1.00|39.00|1.91|
> |+Mercury|0.86|1.24|47.51|1.01|34.65|1.30|
> |+SwiftCoder|0.70|0.95|46.43|1.00|26.02|0.95|

---

### Official Review · Reviewer_jr1T · 2025-03-14

**Overall Recommendation:** 2

**Summary:**

This paper studies the problem of using an LLM to generate higher performance code. The authors propose a pipeline that first constructs a training dataset by sampling the LLM and choosing generations that have higher performance. Then, they finetune the LLM on slow-fast pairs to get it to generate faster code. They evaluate their approach compared to several baselines.

**Claims And Evidence:**

See below.

**Essential References Not Discussed:**

N/A

**Experimental Designs Or Analyses:**

See below.

**Methods And Evaluation Criteria:**

See below.

**Other Comments Or Suggestions:**

N/A

**Other Strengths And Weaknesses:**

Strengths
* Important problem: Code performance is a key problem in programming languages and software engineering research, and LLMs show promise in helping with this problem

Weaknesses
* Unclear novelty: It is not clear what exactly is new about the SwiftCode approach. As far as I understand, they are drawing multiple samples from the LLM, evaluating their performance, and then choosing the best one as the target. There does not appear to be any significant methodological novelty in this pipeline.
* Unclear PIE baseline: The authors compare to PIE, a recent work on performance optimization for C++ programs. However, PIE actually studies several different strategies for using LLMs for code optimization. The authors should clarify which algorithm in the PIE paper they compared against. Several of these techniques studied in that paper involved finetuning and were highly effective.
* Lack of test-time search: I’m wondering why the authors only consider taking a single sample from the LLM at test time (i.e., they only study pass@1). For performance optimization, it is pretty typical to take multiple samples, since we can take the fastest program that passes all the test cases. How does the comparison to baselines scale with the number of samples taken?
* Local execution is noisy: Measuring performance by executing code on a local machine can be very high variance due to a number of factors, including other programs running on the same machine as well as stochasticity in context switches performed by the operating system. The recent PIE paper (Shypula et al., 2024) proposes to use system simulators to mitigate this problem (though it only works for C++, not Python).

**Questions For Authors:**

See above.

**Relation To Broader Scientific Literature:**

See below.

**Theoretical Claims:**

N/A

---

> ### Author Rebuttal · Authors · 2025-04-01
>
> We would like to thank you for your insightful comments and suggestions. We provide detailed responses point by point below. We hope that our clarifications, additional experiments, and responses can address your concerns and lead you to consider increasing your rating of our work.
>
> **W1 Limited novelty**
>
> Our paper's primary contribution is application-driven: we introduce the first multilingual code efficiency instruction tuning dataset. As the ICML guidelines note, "originality need not mean wholly novel methods… a novel dataset... match the needs of the user." Our dataset addresses the critical real-world need for efficient code generation, enabling researchers to fine-tune models for improved performance.
>
> Next, our empirical study provides valuable insights by establishing the correlation between training data efficiency and LLM-generated code efficiency - a relationship not previously well understood in the literature. This finding can inspire future construction of SFT datasets that optimize for both correctness and efficiency. Unlike prior methods, PIE [1], which relies on human programmers to write efficient solutions, or Mercury [2], which selects efficient solutions from pre-existing human-written code, SwiftCoder introduces a fully automated framework that transforms task descriptions into efficient solutions without human intervention.
>
> Finally, SwiftCoder offers greater generalizability by including optimized tasks across 5 programming languages, contrasting with PIE's focus on C++ and Mercury's focus on Python. SwiftCoder's significantly larger scale—comprising 65,710 tasks compared to Mercury's 1,889 and PIE's 1,474—provides more training data, resulting in models that demonstrate superior performance.
>
> [1] Learning performance-improving code edits. ICLR 2024
>
> [2] Mercury: A code efficiency benchmark for code large language models. NeurIPS 2024
>
> **W2 Clarify the method in PIE was compared**
>
> We compared against the best-performed **All** strategy fine-tuned LLM by PIE in the paper. Now, we included comparisons with other PIE strategies in the table below. SwiftCoder outperforms all PIE variants on ET and TMU (e.g., further reduces ET and TMU from **19.6%** and **72.3%** to **22.5%** and **75.9%** for **All** strategy).
>
> |Model|ET|NET|MU|NMU|TMU|NTMU|
> |-|-|-|-|-|-|-|
> |CodeLlama-7B|1.02|0.85|42.33|1.00|23.97|0.82|
> |All|0.82|0.72|8.87|0.18| 6.64|0.24|
> |HQ|1.14|0.98|10.55|0.23|7.06|0.26|
> |All w/Perf-Cond|0.92|0.81|8.91|0.19|6.99|0.25|
> |HQ+Self-Play|0.92|0.80|12.46|0.27|7.80|0.28|
> |SwiftCoder|0.79|0.70|11.06|0.24|5.77|0.21|
>
> **W3 Results with more samples**
>
> Our evaluation follows the setup of prior works (EffiLearner, Mercury), where they use greedy decoding. We conducted additional evaluations for pass@10 (T = 0.8) and will add pass@100 results in our camera-ready manuscript due to rebuttal time limitations. The table below presents our findings across different sampling scenarios. SwiftCoder maintains its efficiency advantage across all configurations.  Even when baselines are sampled 10 times, they still cannot match the efficiency of 1 SwiftCoder sample. When both approaches use the same number of samples, SwiftCoder consistently generates more efficient solutions. Specifically, when both use 10 samples, SwiftCoder improves pass rates by 4-24 percentage points while maintaining better efficiency.
>
> *Reviewer jr1T Ref Table 1*
>
> |Model|ET|NET|MU|NMU|TMU|NTMU|Overlap|Pass@1|
> |-|-|-|-|-|-|-|-|-|
> |**Baseline Sample 1 vs. SwiftCoder sample 1** |
> |deepseek-6.7B| 0.34|2.56|47.26|1.45|30.05|9.97|36.0|44.4|
> |+SwiftCoder| 0.22|1.71|36.31|1.00|9.48|2.11|36.0|51.7|
> |Qwen2.5-7B| 0.31|2.35|31.66|1.00|11.00|2.15|37.2|44.8|
> |+SwiftCoder| 0.16|1.12|31.67|1.00|8.28|1.18|37.2|57.7|
> |**Sample 10 vs. sample 1**|
> |deepseek-6.7B| 0.39|3.00|43.86|1.24|20.14|6.49|48.1|66.8|
> |+SwiftCoder| 0.37|2.68|38.99|1.04|20.13|3.28|48.1|51.9|
> |Qwen2.5-7B| 0.33|2.56|30.99|1.00|9.74|2.33|41.3|49.9|
> |+SwiftCoder| 0.16|1.16|31.01|1.00|7.80|1.24|41.3|57.7|
> |**Sample 10 vs. sample 10**|
> |deepseek-6.7B| 0.44|3.45|41.61|1.19|19.06|6.32|62.5|66.8|
> |+SwiftCoder| 0.40|3.09|42.24|1.19|17.74|5.54|62.5|70.8|
> | Qwen2.5-7B| 0.34|2.59|32.68|1.00|12.09|2.38|48.3|49.9|
> |+SwiftCoder| 0.20|1.49|32.67|1.00|7.82|1.43|48.3|73.8|
>
> **W4 High variance on local machine**
>
> We use an open-sourced code efficiency platform Monolith for evaluation. Similar to the PIE system simulator, Monolith provides consistent performance evaluation but with broader language support. We also report variability results in the Appendix Table 8, where we execute multiple tasks (32 concurrent tasks) from EffiBench 5 different times on Monolith. Our results demonstrate **consistent performance across different execution runs even under concurrent program load, with a coefficient of variation below 3% across all metrics**. This consistency shows that our measurement approach provides reliable results without requiring specialized system simulators.

---

### Decision · Program_Chairs · 2025-05-01

**Decision:**

Accept (poster)

**Comment:**

The paper introduces SWIFTCODE, a framework for enhancing code generation efficiency in LLMs through efficiency-aware fine-tuning. It constructs a training dataset by sampling the LLM and choosing high-performance generations, then fine-tunes the LLM on slow-fast pairs to generate faster code. Evaluations across multiple benchmarks and programming languages demonstrate improved efficiency over baseline methods.

Reviewers acknowledge several strengths of the paper. It addresses an important problem in programming languages and software engineering, filling a gap in code generation research. The development of the SwiftCode dataset is a strength. The dataset is large-scale and comprehensive, covering multiple programming languages and facilitating efficiency-focused fine-tuning. This directly contributes to the framework's ability to enhance code generation efficiency. The experimental evaluation across diverse LLMs and programming languages is extensive. The framework’s simplicity and scalability are also recognized as strengths, with potential applications in software engineering workflows.

On the other hand, the paper also has some weaknesses. The novelty of the approach is limited, as the pipeline closely resembles prior rejection sampling methods in code/math domains. Methodological ambiguities need to be addressed, including the missing correctness verification for candidate solutions, test-time search, and metrics on noisy local execution. There is a lack of clarity regarding the comparison with baselines such as PIE. More in-depth analytical experiments are also expected, such as the exploration of sampling parameters and the generalizability of the trained model.

The authors have submitted their responses to the reviews. Subsequently, all the reviewers have confirmed that they have read the authors' responses, and some of them have updated their overall assessments based on these responses.